# ReSplat: Learning Recurrent Gaussian Splats

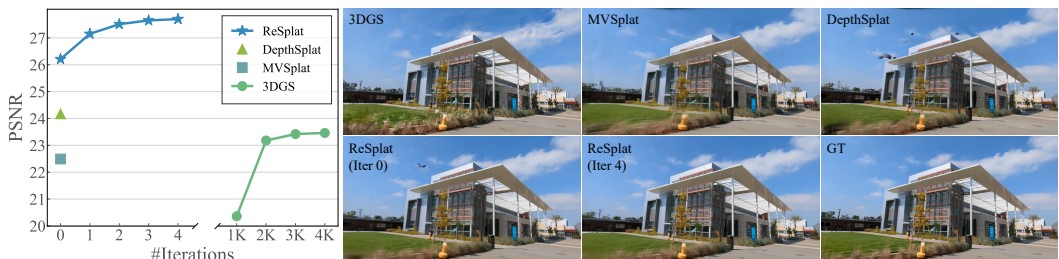

Figure 1: **Learning recurrent Gaussian splats in a feed-forward manner**. We propose ReSplat, a feed-forward recurrent network that iteratively refines 3D Gaussian splats to improve sparse view settings where optimization-based 3DGS (Kerbl et al., 2023) struggles. As initialization (iteration 0), we introduce a compact reconstruction model that predicts Gaussians in a $16\times$ subsampled space, producing $16\times$ fewer Gaussians and $4\times$ faster rendering than per-pixel MVSplat (Chen et al., 2024b) and DepthSplat (Xu et al., 2025). The reduced number of Gaussians makes subsequent refinement efficient. Compared to the optimization-based 3DGS, ReSplat is $100\times$ faster thanks to its feed-forward design, while still benefiting from iterative updates. Here we show results for 8 input views ($512 \times 960$ resolution) on DL3DV dataset; see Table 1 for detailed metrics.

## Abstract

While feed-forward Gaussian splatting models offer computational efficiency and can generalize to sparse input settings, their performance is fundamentally constrained by relying on a single forward pass for inference. We propose ReSplat, a feed-forward recurrent Gaussian splatting model that iteratively refines 3D Gaussians without explicitly computing gradients. Our key insight is that the Gaussian splatting rendering error serves as a rich feedback signal, guiding the recurrent network to learn effective Gaussian updates. This feedback signal naturally adapts to unseen data distributions at test time, enabling robust generalization across datasets, view counts and image resolutions. To initialize the recurrent process, we introduce a compact reconstruction model that operates in a $16\times$ subsampled space, producing $16\times$ fewer Gaussians than previous per-pixel Gaussian models. This substantially reduces computational overhead and allows for efficient Gaussian updates. Extensive experiments across varying of input views (2, 8, 16, 32), resolutions ($256 \times 256$ to $540 \times 960$), and datasets (DL3DV, RealEstate10K and ACID) demonstrate that our method achieves state-of-the-art performance while significantly reducing the number of Gaussians and improving the rendering speed. Our code and models will be public.

## 1 Introduction

Feed-forward Gaussian splatting (Charatan et al., 2024; Szymanowicz et al., 2024) aims to directly predict 3D Gaussian parameters from input images, eliminating the need for expensive per-scene optimization (Kerbl et al., 2023) and enabling high-quality sparse-view reconstruction and view synthesis (Chen et al., 2024b; Liu et al., 2024; Zhang et al., 2024b; Wang et al., 2024). Very recently, significant progress has been made in this line of research: feed-forward models (Zhang et al., 2024a; Xu et al., 2025; Chen et al., 2025b; Ye et al., 2025a; Chen et al., 2024c) can now produce promising reconstruction and view synthesis results from sparse input views.

Despite these advances, the improved performance remains largely concentrated on standard in-domain benchmarks (Zhou et al., 2018; Ling et al., 2023), and existing feed-forward models often

struggle to generalize to new, unseen datasets and scenarios. A primary reason is that most current methods (Charatan et al., 2024; Chen et al., 2024b; Zhang et al., 2024a; Xu et al., 2025; Chen et al., 2025b) focus on learning a single-step mapping from images to 3D Gaussians. While conceptually simple, this approach is inherently limited by the capacity of the employed network, particularly when reconstructing complex and challenging scenes. In contrast, per-scene optimization methods (Kerbl et al., 2023) achieve high-quality results via many iterative updates but are expensive. This motivates our approach: using multiple learned recurrent steps to progressively improve reconstruction quality, balancing the efficiency of feed-forward methods with the adaptability of iterative optimization.

We first identify that the Gaussian splatting rendering error provides a valuable feedback signal informing the model about the quality of its prediction. This also allows the network to adapt to the test data, reducing the dependence on the training distribution and leading to robust generalization. Moreover, this process can be arbitrarily iterated: the model incrementally refines its prediction, improving quality. This recurrent mechanism not only reduces learning difficulty by decomposing the task into smaller, incremental steps, but also increases model expressiveness with each update, approaching the behavior of an infinitely deep network (Bai et al., 2019; Geiping et al., 2025).

Driven by this observation, we begin with a single-step feed-forward Gaussian reconstruction model to initialize the recurrent process, and then perform recurrent updates to improve the initial Gaussians. Since the recurrent updates occur in 3D space, where a large number of Gaussians would impose a significant computational burden, we design our initial model to predict Gaussians in a $16\times$ subsampled space. This contrasts with most existing feed-forward models (Charatan et al., 2024; Szymanowicz et al., 2024; 2025; Zhang et al., 2024a) that predict one or multiple Gaussians per pixel, which scales poorly with increasing numbers of views and image resolutions. Our method achieves a $16\times$ reduction in the number of Gaussians while maintaining performance.

Based on this compact initial reconstruction, we train a weight-sharing recurrent network that iteratively improves the reconstruction. Crucially, the recurrent network leverages the rendering error of input views to determine where and how to update the Gaussians. Specifically, we render the input views (available at test time) using the current prediction, compute the rendering error, and propagate it to 3D Gaussians. The recurrent network then predicts the parameter updates directly from this error and the current Gaussians, without requiring explicit gradient computation.

We validate our method through extensive experiments across diverse scenarios. On the challenging DL3DV (Ling et al., 2023) dataset, using 8 input views at $512 \times 960$ resolution, our learned recurrent model improves PSNR by +3.5dB, while using only $1/16$ of the Gaussians and achieving $4\times$ faster rendering speed. We also demonstrate that our recurrent model leads to robust generalization to unseen datasets, view counts, and image resolutions, where previous single-step feed-forward models usually struggle. With 16 input views at $540 \times 960$ resolution, we outperform Long-LRM (Chen et al., 2025b) by +0.8dB PSNR while using $4\times$ fewer Gaussians. On the commonly used two-view RealEstate10K (Zhou et al., 2018) and ACID (Liu et al., 2021) benchmarks, our ReSplat also achieves state-of-the-art results, demonstrating the strong performance of our method.

## 2 RELATED WORK

**Feed-Forward Gaussian Splatting**. Significant progress has recently been made in feed-forward Gaussian models (Zhang et al., 2024a; Chen et al., 2025b; Xu et al., 2025). However, two major limitations persist: First, most existing feed-forward models predict one or multiple Gaussians for each pixel (Charatan et al., 2024; Szymanowicz et al., 2025; Chen et al., 2025b; 2024b; Zhang et al., 2024a), which will produce millions of Gaussians when handling many input views and/or high-resolution images and thus limit the scalability. Second, most existing methods are developed with single-step feed-forward inference, while conceptually simple, the achievable quality is bounded by the network capability for challenging and complex scenes. In this paper, we overcome these two limitations by first reconstructing Gaussians in a $16\times$ subsampled space, and then performing recurrent Gaussian update based on the rendering error, which significantly improves the efficiency and quality compared to previous methods. Different from the recent method SplatFormer (Chen et al., 2025a) that introduces a single-step refinement network for optimized 3DGS parameters, we propose a weight-sharing recurrent network to iteratively improve the results by using a feed-forward reconstruction as initialization. In addition, SplatFormer is evaluated only on object-centric datasets and it's not straightforward to make it work for complex scenes. In contrast, our ReSplat targets

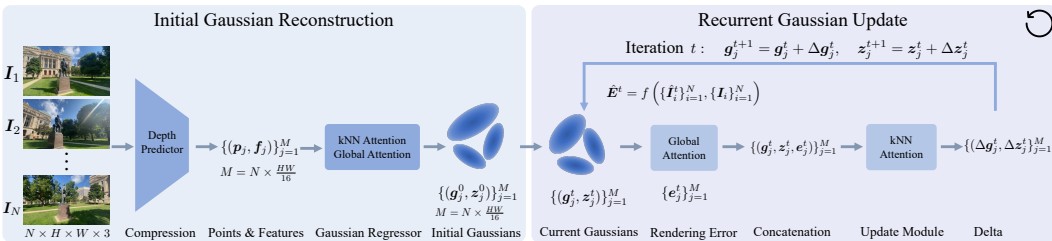

Figure 2: **Learning to recurrently update 3D Gaussians**. Given $N$ posed input images, we first predict per-view depth maps at $1/4$ resolution and then unproject and transform them to a point cloud with image features $\{(\boldsymbol{p}_j, \boldsymbol{f}_j)\}_{j=1}^{M}$, where $M = N \times \frac{HW}{16}$ is the number of points. We then reconstruct an initial set of 3D Gaussians $\{(\boldsymbol{g}_j^0, \boldsymbol{z}_j^0)\}_{j=1}^{M}$ in a $16\times$ subsampled 3D space with a kNN and global attention-based Gaussian regressor. Next, we learn to refine the initial Gaussians recurrently. At each recurrent step $t$, we use the current Gaussian prediction to render input views and then compute the rendering errors $\hat{\boldsymbol{E}}^t$ between rendered and ground-truth input views. A global attention is next applied on the rendering error to propagate the rendering errors to the 3D Gaussians. A kNN attention-based update module next takes as input the concatenation of current Gaussian parameters $\boldsymbol{g}_j^t$, the hidden state $\boldsymbol{z}_j^t$, and the rendering error $\boldsymbol{e}_j^t$, and predicts the incremental updates $\Delta \boldsymbol{g}_j^t$ and $\Delta \boldsymbol{z}_j^t$. We iterate this process until a total number of $T$ steps.

scene-level benchmarks and we demonstrate the effectiveness of the rendering error as an informative feedback signal, which we found crucial but is missing in SplatFormer.

**Learning to Optimize**. Many tasks in machine learning and computer vision can be formulated as a minimization problem with an optimization objective, where the solutions are found by iterative gradient decent (Andrychowicz et al., 2016; Lucas & Kanade, 1981; Sun et al., 2010). Modern approaches (Ma et al., 2020; Teed & Deng, 2020; Metz et al., 2022; Harrison et al., 2022) try to simulate the optimization process by iteratively updating an initial prediction with a weight-sharing network, which usually achieves superior results compared to single-step regression methods, especially for out-of-distribution generalization. In vision, such a framework has been successfully applied to optical flow (Teed & Deng, 2020), stereo matching (Lipson et al., 2021; Wen et al., 2025a), scene flow (Teed & Deng, 2021b), SLAM (Teed & Deng, 2021a), Structure-from-Motion (Li et al., 2025), Multi-View Stereo (Wang et al., 2022), etc. Unlike prior work that often relies on feature correlations (Teed & Deng, 2020) for the recurrent process, we identify the Gaussian rendering error as an informative feedback signal to guide the update of 3D Gaussian parameters.

**Learning to Optimize for View Synthesis**. In the context of view synthesis, DeepView (Flynn et al., 2019) predicts multi-plane images with learned gradient decent, where explicit gradient computation is necessary. In addition, G3R (Chen et al., 2024d) proposes to learn to iteratively refine the 3D Gaussians with the guidance of the explicitly computed gradients. However, our method is gradient free. Moreover, G3R requires well-covered 3D points for initialization and struggles with sparse points, while we directly predict initial Gaussians from posed images, without requiring any initial 3D points. Similar to G3R, QuickSplat (Liu et al., 2025) also relies on gradient computation but focuses on surface reconstruction. Another related work LIFe-GOM (Wen et al., 2025b) tries to iteratively update the 3D representation in a gradient-free manner, but it focuses on the task of human avatars with a hybrid Gaussian-mesh 3D representation. In contrast, our method aims to improve the quality and generalization of feed-forward Gaussian splatting models for general scenes.

## 3 APPROACH

Given $N$ input images $\{\boldsymbol{I}^i\}_{i=1}^{N} (\boldsymbol{I}^i \in \mathbb{R}^{H \times W \times 3})$ with their intrinsic $\{\boldsymbol{K}^i\}_{i=1}^{N} (\boldsymbol{K}^i \in \mathbb{R}^{3 \times 3})$ and extrinsic $\{(\boldsymbol{R}_i, \boldsymbol{t}_i)\}_{i=1}^{N} (\boldsymbol{R}_i \in \mathrm{SO}(3), \boldsymbol{t}_i \in \mathbb{R}^3)$ matrices, our goal is to predict a set of 3D Gaussian primitives (Kerbl et al., 2023) $\mathcal{G} = \{(\boldsymbol{\mu}_j, \alpha_j, \boldsymbol{\Sigma}_j, \mathbf{sh}_j)\}_{j=1}^{M}$ to model the scene, where $M$ is the total number of Gaussian primitives and $\boldsymbol{\mu}_j, \alpha_j, \boldsymbol{\Sigma}_j$ and $\mathbf{sh}_j$ are the 3D Gaussian's position, opacity, covariance, and spherical harmonics, respectively. The reconstructed 3D Gaussians can be efficiently rasterized, enabling fast and high-quality novel view synthesis.

Unlike previous feed-forward models (Charatan et al., 2024; Zhang et al., 2024a; Chen et al., 2025b) that perform a single-step feed-forward prediction, we propose to learn to estimate the Gaussian parameters recurrently. This not only reduces learning difficulty by decomposing the task into smaller, incremental steps but also enables higher reconstruction quality. In particular, we first predict an initial set of 3D Gaussians and then iteratively refine them in a gradient-free, feed-forward manner. Given that the Gaussian update occurs in the 3D space, a large number of 3D Gaussians will introduce significant computational overhead during the update process. Thus, in our initial reconstruction stage, we predict a compact set of 3D Gaussians in a $16\times$ subsampled space. More specifically, we perform $4\times$ spatial compression when predicting per-view depth maps which leads to $16\times$ fewer number of Gaussians compared to previous per-pixel representation (Charatan et al., 2024; Szymanowicz et al., 2024). Thus, the number of Gaussians $M$ in our model is $N \times \frac{HW}{16}$, which scales efficiently to many input views and high-resolution images. Fig. 2 provides an overview of our pipeline.

## 3.1 INITIAL GAUSSIAN RECONSTRUCTION

**Subsampled 3D Space.** Our initial Gaussian reconstruction model is based on the DepthSplat (Xu et al., 2025) architecture. However, unlike DepthSplat, we predict Gaussians in a spatially $16\times$ subsampled 3D space ($N \times \frac{HW}{16}$), and thus our number of Gaussians is $16\times$ fewer than DepthSplat. To achieve $16\times$ subsampling, we resize the full resolution depth predictions from the depth model in DepthSplat to $1/4$ resolution ($N \times \frac{H}{4} \times \frac{W}{4}$), and then unproject and transform them in 3D via camera parameters to obtain a point cloud with $M = N \times \frac{HW}{16}$ points. Each 3D point $\boldsymbol{p}_j \in \mathbb{R}^3$ is also associated with a feature vector $\boldsymbol{f}_j \in \mathbb{R}^{C_1}$ extracted from the input images:

$$\{\boldsymbol{I}_i, \boldsymbol{K}_i, \boldsymbol{R}_i, \boldsymbol{t}_i\}_{i=1}^N \to \{(\boldsymbol{p}_j, \boldsymbol{f}_j)\}_{j=1}^M. \tag{1}$$

Since we now have $16\times$ fewer 3D points, naïvely predicting Gaussian parameters from the point features $\boldsymbol{f}_j$ will lead to considerable performance loss. However, we found that using additional kNN attention (Zhao et al., 2021) and global attention (Vaswani et al., 2017) layers on the point cloud to encode the 3D context (Xu et al., 2024; Chen et al., 2024a) information can compensate for this loss.

**Aggregating the 3D Context.** We use six alternative blocks of kNN attention and global attention to model both local and global 3D contexts, which enables communication between different 3D points and produce 3D context-aggregated features $\boldsymbol{f}_j^* \in \mathbb{R}^{C_1}$ with increased expressiveness:

$$\{(\boldsymbol{p}_j, \boldsymbol{f}_j)\}_{j=1}^M \to \{(\boldsymbol{p}_j, \boldsymbol{f}_j^*)\}_{j=1}^M. \tag{2}$$

**Decoding to Gaussians.** We use the point cloud $\{\boldsymbol{p}_j\}_{j=1}^M$ as the Gaussian centers and other Gaussian parameters are decoded with a lightweight Gaussian head (two-layer MLP) from the 3D context-aggregated features $\{\boldsymbol{f}_j^*\}_{j=1}^M$. Accordingly, we obtain an initial set of 3D Gaussians with parameters $\{(\boldsymbol{\mu}_j, \alpha_j, \boldsymbol{\Sigma}_j, \mathbf{sh}_j)\}_{j=1}^M$ and feature vectors $\{\boldsymbol{f}_j^*\}_{j=1}^M$. We use $\boldsymbol{g}_j^0 \in \mathbb{R}^{C_2}$ to denote the concatenation of all the Gaussian parameters $(\boldsymbol{\mu}_j, \alpha_j, \boldsymbol{\Sigma}_j, \mathbf{sh}_j)$ for the $j$-th Gaussian at initialization, where $C_2$ is the total number of parameters for each Gaussian. We use $\boldsymbol{z}_j^0$ to denote the initial hidden state of the $j$-th Gaussian for the subsequent recurrent process, and initialize it with feature $\boldsymbol{f}_j^*$: $\boldsymbol{z}_j^0 = \boldsymbol{f}_j^* \in \mathbb{R}^{C_1}$. Thus, the initial Gaussians can be represented as

$$\mathcal{G}^0 = \{(\boldsymbol{g}_j^0, \boldsymbol{z}_j^0)\}_{j=1}^M. \tag{3}$$

## 3.2 RECURRENT GAUSSIAN UPDATE

Based on the initial Gaussian prediction in Section 3.1 (Eq. (3)), we train a recurrent network which iteratively refines the initial prediction. In particular, at iteration $t$, ($t = 0, 1, \cdots, T-1$, $T$ is the total number of iterations), the recurrent network predicts incremental updates to all Gaussian parameters $\Delta \boldsymbol{g}_j^t \in \mathbb{R}^{C_2}$ and their hidden state $\Delta \boldsymbol{z}_j^t \in \mathbb{R}^{C_1}$ as:

$$\boldsymbol{g}_j^{t+1} = \boldsymbol{g}_j^t + \Delta \boldsymbol{g}_j^t, \quad \boldsymbol{z}_j^{t+1} = \boldsymbol{z}_j^t + \Delta \boldsymbol{z}_j^t. \tag{4}$$

To predict the incremental updates $\Delta \boldsymbol{g}_j^t$ and $\Delta \boldsymbol{z}_j^t$, we propose to learn the update in a gradient-free, feed-forward manner from the rendering error of input views.

**Computing the Rendering Error**. Given that we have access to the input views at test time, we are able to create a feedback loop to guide the recurrent network to learn the incremental updates. More specifically, we first render the input views $\{\hat{I}_i^t\}_{i=1}^N$ based on the current Gaussian parameters at iteration $t$, and then measure the difference between the rendered and ground-truth input views. We evaluate several different methods to compute the rendering error and observe that a combination of pixel-space and feature-space rendering errors performs best.

In particular, we first use $\{\hat{I}_i^t - I_i\}_{i=1}^N$ to measure the rendering error in the pixel space, and then perform $4\times$ spatial downsampling with pixel unshuffle to align with the number of 3D Gaussians. For the feature-space rendering error, we extract the first-three-stage features (at $1/2$, $1/4$ and $1/8$ resolutions) of the ImageNet (Deng et al., 2009) pre-trained ResNet-18 (He et al., 2016) for the rendered input views and ground-truth input views, and bilinearly resize the three-scale features to the same $1/4$ resolution, followed by concatenation. We denote the extracted features as $\{\hat{F}_i^t\}_{i=1}^N$ and $\{F_i\}_{i=1}^N$ ($\hat{F}_i^t, F_i \in \mathbb{R}^{\frac{H}{4} \times \frac{W}{4} \times C_3}$) for rendered and ground-truth input views, respectively. We then compute the difference between features with subtraction $\{\hat{F}_i^t - F_i\}_{i=1}^N$. We combine pixel-space and feature-space rendering errors via element-wise addition. To match channel dimensions, the pixel-space error is first projected to the feature space using a linear layer followed by LayerNorm (Ba et al., 2016). This process can be expressed as

$$\hat{E}^t = f\left(\{\hat{I}_i^t\}_{i=1}^N, \{I_i\}_{i=1}^N\right) = \{\hat{F}_i^t - F_i\}_{i=1}^N + \text{proj}(\{\hat{I}_i^t - I_i\}_{i=1}^N), \tag{5}$$

where "proj" is the operation mentioned before to match dimensions. We denote all rendering errors as $\hat{E}^t = \{\hat{e}_j^t\}_{j=1}^{N \times \frac{H}{4} \times \frac{W}{4}}$, where $\hat{e}_j^t \in \mathbb{R}^{C_3}$ is the $j$-th feature difference of dimension $C_3$ at iteration $t$.

**Propagating the Rendering Error to Gaussians**. To propagate the rendering error to 3D Gaussians such that they can guide the network to update the Gaussians. A straightforward approach is to concatenate the rendering error $\hat{e}_j^t$ with the Gaussians $(g_j^t, z_j^t)$ in a spatially aligned manner since they have the same number of points ($N \times \frac{HW}{16}$). However, in this way, the $j$-th Gaussian can only receive local information around the $j$-th rendered pixel, while it can also contribute to other rendered pixels during the rendering process. To propagate the rendering error more effectively, we propose to apply global attention on all the $N \times \frac{H}{4} \times \frac{W}{4}$ rendering errors $\hat{E}^t$, which enables each Gaussian to receive information from all rendering errors. This process can be formulated as:

$$E^t = \text{global\_attention}(\hat{E}^t) = \{e_j^t\}_{j=1}^{N \times \frac{HW}{16}}, \tag{6}$$

where $e_j^t$ is the $j$-th rendering error which has aggregated the original point-wise rendering error $\hat{e}_j^t$ globally. We then concatenate the Gaussians with the globally aggregated rendering errors as $\{(g_j^t, z_j^t, e_j^t)\}_{j=1}^M$, which are next used to predict the incremental update (illustrated in Fig. 2).

**Recurrent Gaussian Update**. Let the Gaussians at iteration $t$ be $\mathcal{G}^t = \{(g_j^t, z_j^t)\}_{j=1}^M$, our update module predicts the incremental updates of Gaussian parameters and hidden state as:

$$\{(g_j^t, z_j^t, e_j^t)\}_{j=1}^M \rightarrow \{(\Delta g_j^t, \Delta z_j^t\}_{j=1}^M, \tag{7}$$

and then they are added to the current prediction (Eq. (4)). This process is iterated $T$ times. We observe that our model converges after 4 iterations. During training, we randomly sample the number of iterations $T$ between 1 and 4, and our model supports different number of iterations at inference time, allowing a flexible trade-off between accuracy and speed with a single model. Since the recurrent process occurs in the 3D space, we choose to use four kNN attention (Zhao et al., 2021) blocks as the recurrent architecture to model the local structural details. The Gaussian updates $g_j^t$ are decoded with a lightweight update head (four-layer MLP).

## 3.3 TRAINING LOSS

Our model is trained in two stages. In the first stage, we train an initial Gaussian reconstruction model to provide compact initializations to our subsequent updates. The training loss is a combination of a rendering loss $\ell_{\text{render}}$ and an edge-aware depth smoothness loss $\ell_{\text{depth\_smooth}}$ on the predicted depth

maps of the input views:

$$L_{1\text{st}} = \sum_{v=1}^{V} \ell_{\text{render}}(\hat{\boldsymbol{I}}_v, \boldsymbol{I}_v) + \alpha \cdot \sum_{i=1}^{N} \ell_{\text{depth\_smooth}}(\boldsymbol{I}_i, \hat{\boldsymbol{D}}_i), \qquad (8)$$

$$\ell_{\text{render}}(\hat{\boldsymbol{I}}, \boldsymbol{I}) = \ell_1(\hat{\boldsymbol{I}}, \boldsymbol{I}) + \lambda \cdot \ell_{\text{perceptual}}(\hat{\boldsymbol{I}}, \boldsymbol{I}), \qquad (9)$$

$$\ell_{\text{depth\_smooth}}(\boldsymbol{I}, \hat{\boldsymbol{D}}) = |\partial_x \hat{\boldsymbol{D}}| e^{-|\partial_x \boldsymbol{I}|} + |\partial_y \hat{\boldsymbol{D}}| e^{-|\partial_y \boldsymbol{I}|}. \qquad (10)$$

where $V$ is the number of target views to render in each training step. $N$ is the number of input views. The perceptual loss $\ell_{\text{perceptual}}$ (Johnson et al., 2016) measures the distance in the VGG (Simonyan & Zisserman, 2014)'s feature space, which is also used in previous methods (Zhang et al., 2024a; Jin et al., 2025). The depth smoothness loss $\ell_{\text{depth\_smooth}}$ is a regularization term on the estimated depth maps of the input views to encourage the depth gradient to be similar to the image gradient (Godard et al., 2017; 2019). We use $\alpha = 0.01$ and $\lambda = 0.5$ for all the experiments.

In the second stage, we freeze our initial reconstruction model and train only the recurrent model end-to-end. We use the rendering loss $\ell_{\text{render}}$ of rendered and ground truth target views to supervise the network. All the Gaussian predictions during the recurrent process are supervised with the rendering loss with exponentially ($\gamma = 0.9$) increasing weights:

$$L_{2\text{nd}} = \sum_{t=0}^{T-1} \gamma^{T-1-t} \sum_{v=1}^{V} \ell_{\text{render}}(\hat{\boldsymbol{I}}_v^t, \boldsymbol{I}_v). \qquad (11)$$

## 4 EXPERIMENTS

**Implementation Details.** We implement our method in PyTorch and use Flash Attention 3 (Shah et al., 2024) for efficient attention computations. We choose $k = 16$ for kNN attentions following Point Transformer (Zhao et al., 2021), and our Gaussian splatting renderer is based on gsplat (Ye et al., 2025b)'s Mip-Splatting (Yu et al., 2024) implementation. We optimize our model with AdamW (Loshchilov & Hutter, 2017) optimizer. More training details are presented in the appendix.

Our model contains several global attention layers. Considering that performing global attention on $N \times \frac{H}{4} \times \frac{W}{4}$ features would be expensive for high-resolution images, we first perform $4\times$ spatial downsampling with pixel unshuffle (reshaping from the spatial dimension to the channel dimension) and then compute global attention on the $N \times \frac{H}{16} \times \frac{W}{16}$ features. Finally, we upsample the features back to the $1/4$ resolution with pixel shuffle (reshaping from the channel dimension to the spatial dimension). This implementation enables our model to scale efficiently to high-resolution images.

Our default model uses a ViT-B (Dosovitskiy et al., 2020) backbone as part of our depth prediction model, which has 223M parameters in total (209M for the initialization model and 14M for the recurrent model). For ablation experiments, we use a ViT-S backbone to save compute, which has 76M parameters in total (62M for the initialization model and 14M for the recurrent model).

We will release our code, pre-trained models, training and evaluation scripts to ease reproducibility.

**Coordinate System.** Since our recurrent network operates within a global 3D space, the selection of a coordinate system is critical, as it directly determines the spatial distribution of the Gaussian's centers. Our datasets consist of video sequences with camera poses estimated from COLMAP (Schonberger & Frahm, 2016). We evaluated aligning the global reference frame to the first, middle, and last views of the input images. Empirically, we observed that using the middle input view as the reference coordinate system yields the best performance (see Table 6b). We posit that this centers the coordinate system, reducing the maximum transformation distance to the first and last input views and effectively balancing the spatial positions of the 3D Gaussians.

**Evaluation Setup.** We mainly consider three evaluation setups. First, view synthesis from 8 input views at $512 \times 960$ resolution on DL3DV, where we re-train 3DGS (Kerbl et al., 2023), MVSplat (Chen et al., 2024b) and DepthSplat (Xu et al., 2025) with their public code for fair comparisons. Second, view synthesis from 16 input views at $540 \times 960$ resolution on DL3DV following Long-LRM (Chen et al., 2025b), with which we could perform comparisons with Long-LRM since its model weights are not released and re-training would be expensive. Third, we also evaluate on the commonly used 2-view ($256 \times 256$) setup on RealEstate10K and ACID, where we compare with related methods like GS-LRM (Zhang et al., 2024a) and LVSM (Jin et al., 2025).

Table 1: **Evaluation of 8 input views** ($512 \times 960$) **on DL3DV**. The standard optimization-based approach 3DGS (Kerbl et al., 2023) requires several thousands of iterations to reach convergence, while our feed-forward ReSplat is significantly faster and is able to benefit from additional iterations. Previous per-pixel feed-forward models MVSplat (Chen et al., 2024b) and DepthSplat (Xu et al., 2025) produces millions of Gaussians, while our ReSplat compresses the number of Gaussians by $16\times$, also leading to $4\times$ faster rendering speed.

| Method | Category | #Iterations | PSNR ↑ | SSIM ↑ | LPIPS ↓ | #Gaussians | Recon. Time (s) | Render Time (s) |
|---|---|---|---|---|---|---|---|---|
| 3DGS | Optimization | 1000 | 20.36 | 0.667 | 0.448 | 9K | 15 | 0.0001 |
| | | 2000 | 23.18 | 0.763 | 0.269 | 137K | 31 | 0.0005 |
| | | 3000 | 23.42 | 0.770 | 0.232 | 283K | 50 | 0.0008 |
| | | 4000 | 23.46 | 0.770 | 0.224 | 359K | 70 | 0.0009 |
| MVSplat | Feed-Forward | 0 | 22.49 | 0.764 | 0.261 | 3932K | **0.129** | 0.0030 |
| DepthSplat | Feed-Forward | 0 | 24.17 | 0.815 | 0.208 | 3932K | 0.190 | 0.0030 |
| ReSplat | Feed-Forward | 0 | 26.21 | 0.842 | 0.185 | **246K** | 0.311 | **0.0007** |
| | | 1 | 27.15 | 0.859 | 0.169 | **246K** | 0.437 | **0.0007** |
| | | 2 | 27.51 | 0.865 | 0.163 | **246K** | 0.563 | **0.0007** |
| | | 3 | 27.65 | 0.867 | 0.161 | **246K** | 0.789 | **0.0007** |
| | | 4 | **27.70** | **0.868** | **0.160** | **246K** | 0.816 | **0.0007** |

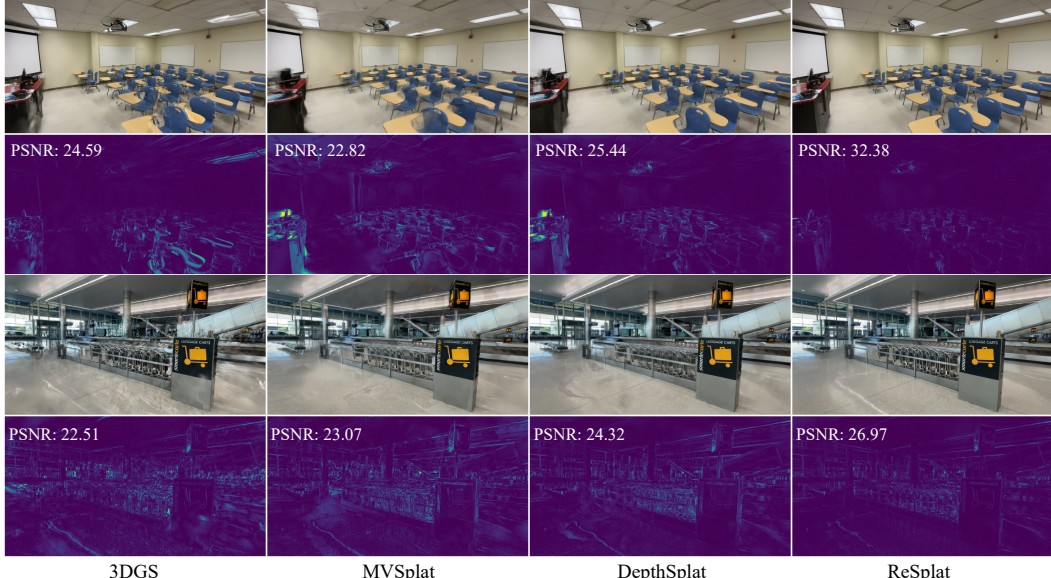

Figure 3: **View synthesis on DL3DV**. Our ReSplat outperforms both optimization and feed-forward methods, with significantly smaller rendering errors. More samples are presented in Fig. 8 (appendix).

## 4.1 MAIN RESULTS

**8 Views at** $512 \times 960$ **Resolution on DL3DV.** We report the results on the DL3DV benchmark split (140 scenes) in Table 1. For experiments with 3DGS (Kerbl et al., 2023), we perform per-scene optimization on the 8 input views for all the 140 scenes, while for feed-forward models, we perform zero-shot feed-forward inference. We observe that 3DGS optimization typically converges with 4K optimization steps, and optimizing longer could lead to overfitting due to the sparse input views, thus we report the best results at 4K iterations. As shown in Table 1, 3DGS optimization is considerably slow due to the large number of iterations required, while our feed-forward ReSplat is $100\times$ faster and is able to benefit from additional iterations. Previous per-pixel feed-forward models MVSplat (Chen et al., 2024b) and DepthSplat (Xu et al., 2025) produces millions of Gaussians, while our ReSplat compresses the number of Gaussians by $16\times$, also leading to $4\times$ faster rendering speed. Overall, our ReSplat outperforms 3DGS by 4.2dB PSNR and DepthSplat by 3.5dB PSNR with superior efficiency on the number of Gaussians and the rendering speed. The visual comparisons are shown in Fig. 3 and Fig. 8, which demonstrate the higher rendering quality of our method. We provide additional visualizations of reconstructed Gaussians in Fig. 12 and Fig. 13 (appendix).

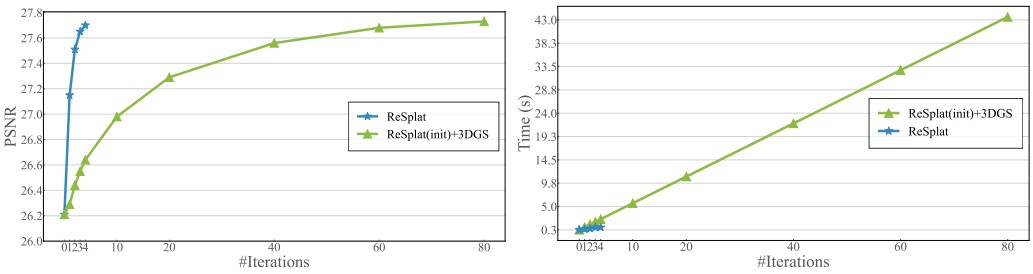

(a) **PSNR *vs*. number of iterations.**  (b) **Reconstruction time *vs*. number of iterations.**

Figure 4: **Optimization-based *vs*. feed-forward refinement.** Starting from the same ReSplat's initialization, we compare 3DGS optimization-based refinement with our feed-forward approach. Our ReSplat improves the rendering quality significantly faster (4 *vs*. 80 iterations) and is $53\times$ faster in terms of the reconstruction speed. See Table 8 (appendix) for detailed numbers.

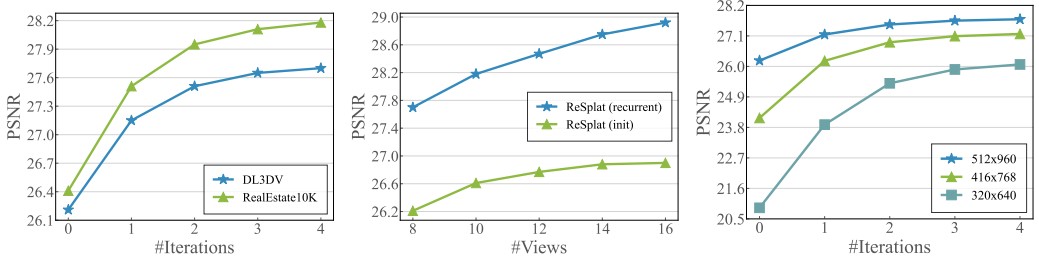

(a) **Cross-dataset generalization.**  (b) **Cross-view generalization.**  (c) **Cross-resolution generalization.**

Figure 5: **Generalization to unseen datasets, number of input views, and image resolutions.** Our recurrent model demonstrates robust generalization despite being trained exclusively on DL3DV at a fixed resolution ($512 \times 960$) with 8 input views. We note that the RealEstate10K dataset is easier than DL3DV due to smaller camera motion.

**Optimization-Based *vs*. Feed-Forward Refinement.** To further demonstrate the superiority of our feed-forward refinement, we compare with optimization-based 3DGS by using the same ReSplat's initialization. From Fig. 4 and Table 8 (appendix), we can observe that our ReSplat is significantly faster than 3DGS optimization-based refinement thanks to our gradient-free and feed-forward nature.

**Generalization Across Datasets, View Counts, and Image Resolutions.** We further evaluate the generalization abilities of our feed-forward model trained on exclusively DL3DV at $512 \times 960$ resolution with 8 input views. First, when generalizing to the unseen RealEstate10K dataset (Fig. 5a), the improvement brought by our recurrent model is more significant since unlike single-step feed-forward models (iteration 0), our model is able to adapt to the test data using the rendering error and thus reduces the domain gap. Second, we test our initial and recurrent models with more than 8 input views in Fig. 5b, and observe that our recurrent model benefits more from the additional input views, while the initial model tends to be saturated. This indicates that our rendering error-informed recurrent model exploits the additional information more effectively. Third, existing single-step feed-forward models usually suffer from obvious performance drop when the testing image resolution is different from training. However, with our recurrent model, we can significantly improve the robustness to different testing resolutions (Fig. 5c). For example, our recurrent model improves 5dB PSNR when generalizing from $512 \times 960$ to $320 \times 640$. These experiments demonstrate the benefits of our recurrent model, which is able to adapt to the unseen test scenario using the rendering error as a feedback signal and thus substantially improves the robustness.

**16 Views at** $540 \times 960$ **Resolution on DL3DV.** We follow Long-LRM (Chen et al., 2025b) for this evaluation setup such that a direct comparison is possible. The results of 3DGS (Kerbl et al., 2023), Mip-Splatting (Yu et al., 2024), and Scaffold-GS (Lu et al., 2024) are borrowed from Long-LRM paper. This experiment aims to reconstruct the full DL3DV scene from 16 input views, which is particularly challenging due to the wide coverage of the DL3DV dataset. However, our ReSplat still outperforms previous optimization and feed-forward methods, as shown in Table 2. Notably, Long-LRM uses Gaussian pruning based on the opacity values during training and evaluation, which

Table 2: **Evaluation of 16 input views** ($540 \times 960$) **on DL3DV.** Our ReSplat reconstructs $4\times$ fewer Gaussians than Long-LRM but still outperforms it.

| Method | #Iterations | PSNR ↑ | SSIM ↑ | LPIPS ↓ | Recon. Time | #Gaussians |
|---|---|---|---|---|---|---|
| 3DGS (Kerbl et al., 2023) | 30000 | 21.20 | 0.708 | 0.264 | 13min | - |
| Mip-Splatting (Yu et al., 2024) | 30000 | 20.88 | 0.712 | 0.274 | 13min | - |
| Scaffold-GS (Lu et al., 2024) | 30000 | 22.13 | 0.738 | **0.250** | 16min | - |
| Long-LRM (Chen et al., 2025b) | 0 | 22.66 | 0.740 | 0.292 | **0.4sec** | 2073K |
| | 0 | 22.69 | 0.742 | 0.307 | 0.7sec | **518K** |
| ReSplat | 1 | 23.23 | 0.758 | 0.291 | 1.2sec | **518K** |
| | 2 | **23.51** | **0.766** | 0.284 | 1.7sec | **518K** |

| Method | w/ 3DGS | PSNR ↑ | SSIM ↑ | LPIPS ↓ |
|---|---|---|---|---|
| pixelSplat | ✓ | 25.89 | 0.858 | 0.142 |
| MVSplat | ✓ | 26.39 | 0.869 | 0.128 |
| DepthSplat | ✓ | 27.47 | 0.889 | 0.114 |
| GS-LRM | ✓ | 28.10 | 0.892 | 0.114 |
| Long-LRM | ✓ | 28.54 | 0.895 | 0.109 |
| LVSM (enc-dec) | ✗ | 28.58 | 0.893 | 0.114 |
| LVSM (dec-only) | ✗ | 29.67 | 0.906 | **0.098** |
| ReSplat (Ours) | ✓ | **29.75** | **0.912** | 0.100 |

Table 3: **Evaluation of two input views** ($256 \times 256$) **on RealEstate10K**. ReSplat outperforms prior feed-forward 3DGS models and matches LVSM's quality with $20\times$ faster rendering.

Figure 6: **Visual comparisons on RealEstate10K**. ReSplat produces sharper structures than MVSplat and DepthSplat.

Table 4: **RealEstate10K to ACID generalization**.

| Method | PSNR ↑ | SSIM ↑ | LPIPS ↓ |
|---|---|---|---|
| pixelSplat | 27.64 | 0.830 | 0.160 |
| MVSplat | 28.15 | 0.841 | 0.147 |
| DepthSplat | 28.37 | 0.847 | 0.141 |
| GS-LRM | 28.84 | 0.849 | 0.146 |
| ReSplat | **29.87** | **0.864** | **0.135** |

Table 5: **Different compression factors**. $16\times$ compression represents a good speed-accuracy trade-off. The visual results are shown in Fig. 7.

| Compression | PSNR ↑ | SSIM ↑ | LPIPS ↓ | Time (s) |
|---|---|---|---|---|
| $64\times$ | 24.77 | 0.797 | 0.226 | **0.096** |
| $16\times$ | 26.77 | 0.865 | 0.142 | 0.104 |
| $4\times$ | **28.36** | **0.900** | **0.103** | 0.206 |

leads to $\sim 4\times$ reduction of number of Gaussians. In contract, we compress the Gaussians by $16\times$, thus our final reconstruction has $4\times$ fewer Gaussians than Long-LRM while still outperforming it. Our reconstruction time is slower than Long-LRM mainly because of the kNN operation, which is expensive to compute with large number (*e.g.*, $> 500$K) of points. Further optimizing the implementation could potentially improve our reconstruction speed.

**2 Views at** $256 \times 256$ **Resolution on RealEstate10K and ACID**. Considering the redundancy is not a major issue with two input views at low resolution ($256 \times 256$), we perform $4\times$ subsampling in the 3D space and decode 4 Gaussians from each subsampled 3D point in our initial reconstruction model. Thus the number of Gaussians remains the same as previous per-pixel methods. The recurrent process remains the same as the many-view setups. Table 3 shows that our ReSplat outperforms previous feed-forward 3DGS models (*e.g.*, DepthSplat (Xu et al., 2025), GS-LRM (Zhang et al., 2024a) and Long-LRM (Chen et al., 2025b)) by clear margins. Compared to the 3DGS-free method LVSM (Jin et al., 2025), we outperform its encoder-decoder architecture by 1.1dB PSNR and our results are similar to its best-performing decoder-only model variant. However, our method offers the benefits of an explicit 3D Gaussian representation, enabling $20\times$ faster rendering speed. We present visual comparisons in Fig. 6, where our ReSplat produces better structures than MVSplat and DepthSplat. In Table 4, we show the zero-shot generalization results on the unseen ACID dataset, where our ReSplat again outperforms previous methods by clear margins.

## 4.2 ANALYSIS AND ABLATION

We conduct several experiments to analyze the properties of our model and validate our design choices. To save compute, all experiments in this section are performed using 8 input views at $256 \times 448$ resolution on the DL3DV dataset, and we train all the models on 4 GH200 GPUs.

Table 6: **Analysis and Ablation**. We conduct several experiments to validate our design choices.

(a) **Ablation of the rendering error**.

| Method | PSNR ↑ | SSIM ↑ | LPIPS ↓ |
|---|---|---|---|
| Initialization | 26.77 | 0.865 | 0.142 |
| w/o rendering error | 27.19 | 0.873 | 0.137 |
| RGB error only | 27.90 | 0.882 | 0.130 |
| Feature error only | 28.77 | 0.897 | 0.110 |
| Concat (RGB & feature errors) | 28.93 | 0.900 | 0.106 |
| Add (RGB & feature errors) | **29.07** | **0.902** | **0.105** |

(b) **Ablation of the coordinate system**.

| Reference | PSNR ↑ | SSIM ↑ | LPIPS ↓ |
|---|---|---|---|
| Initialization | 26.77 | 0.865 | 0.142 |
| COLMAP | 28.14 | 0.886 | 0.116 |
| First view | 28.66 | 0.896 | 0.109 |
| Last view | 28.59 | 0.895 | 0.110 |
| Middle view | **29.07** | **0.902** | **0.105** |

(c) **Ablation of the initial reconstruction model**.

| Method | PSNR ↑ | SSIM ↑ | LPIPS ↓ | #Gaussians |
|---|---|---|---|---|
| DepthSplat | 25.79 | 0.861 | **0.134** | 918K |
| Full | **26.77** | **0.865** | 0.142 | **57K** |
| w/o kNN attn | 25.30 | 0.833 | 0.178 | **57K** |
| w/o global attn | 26.33 | 0.856 | 0.150 | **57K** |
| w/o kNN, w/o global | 24.50 | 0.814 | 0.200 | **57K** |

(d) **Ablation of the recurrent model**.

| Method | PSNR ↑ | SSIM ↑ | LPIPS ↓ |
|---|---|---|---|
| Initialization | 26.77 | 0.865 | 0.142 |
| Full | **29.07** | **0.902** | **0.105** |
| w/o state | 27.79 | 0.878 | 0.125 |
| w/o kNN attn | 28.58 | 0.894 | 0.111 |
| w/o global attn | 28.96 | 0.900 | 0.107 |

**Rendering Error.** We evaluate the effect of the rendering error in Table 6a. Removing the rendering error in our recurrent model leads to a significant performance drop (-1.9dB PSNR). The feature-space error performs better than the pixel-space RGB error, and the best performance is obtained by combining the RGB-space and feature-space errors (addition is slightly better than concatenation).

**Coordinate System.** We observe that aligning to the middle input view's camera pose performs significantly better (+0.9dB PSNR) than using the default COLMAP's global coordinate system. We attribute this to the temporal nature of video data, where anchoring to the middle frame acts as a central pivot, balancing the spatial distributions of the 3D Gaussians and making the learning task easier for the 3D network.

**Compression Factor**. Be default, we compress the number of Gaussians by $16\times$ using depth maps at $1/4$ resolution. In Table 5, we compare with $64\times$ (with $1/8$ depth maps) and $4\times$ (with $1/2$ depth maps) compression factors. We can observe that less compression leads to higher quality, as also visualized in Fig. 7 (appendix). However, $4\times$ compression is $2\times$ slower than $16\times$ for 8 input views at $256 \times 448$ resolution. It would be more expensive when handling higher resolution images (*e.g.*, $512 \times 960$). Thus, we choose $16\times$ compression as a good speed-accuracy trade-off.

**Ablation of the Initial Model.** As shown in Table 6c, we observe that the kNN attention is crucial to maintain our performance when compressing the number of Gaussians by $16\times$. The global attention also brings moderate performance gains, which indicates that both local and global 3D contexts are important for learning compact 3D representations. The visual results are in Fig. 10 (appendix).

**Ablation of the Recurrent Model.** The state input is pivotal to our recurrent network. Unlike the low-level, raw Gaussian parameters, it encodes high-level image and 3D features derived from our initialization model. Both kNN attention and global attention contribute to the performance. The visual results are in Fig. 11 (appendix).

## 5 CONCLUSION

We have presented a feed-forward recurrent Gaussian splatting model that achieves efficient and high-quality view synthesis. By using the rendering error as an informative feedback signal and operating in a compact subsampled 3D space, our method significantly reduces the number of Gaussians while improving performance and generalization across datasets, view counts, and image resolutions.

**Limitations**. Our current model relies on kNN-based point attention (Zhao et al., 2021), which incurs high computational cost when the number of Gaussians becomes large (*e.g.*, > 500K). We anticipate that more efficient point-based attentions (Wu et al., 2022; 2024) and sparse structures (Ren et al., 2024) could further improve the scalability and efficiency of our approach. In addition, our current model saturates after four iterations, it would be interesting to explore how to further scale up test-time compute. We view these directions as promising avenues for future research.

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

## A  ADDITIONAL EVALUATIONS

**Comparison with 3DGS Across 8, 16, and 32 Views.** In Table 7, we compare with optimization-based 3DGS across 8, 16 and 32 input views. The quality gap becomes smaller when given 32 views for optimization-based 3DGS. However, our ReSplat is more than $200\times$ faster in terms of the reconstruction speed. In this setup, we expand the sampling region as the number of input views increases to enlarge scene coverage. Consequently, the test views differ across configurations to account for this larger spatial extent.

**Optimization-Based *vs*. Feed-Forward Refinement.** In Table 8, we show the detailed numbers of comparing 3DGS optimization-based and our feed-forward refinement starting from the same ReSplat's initialization. See Fig. 4 (main paper) for the visualization of this table.

**Recurrent *vs*. Non-recurrent Architecture.** We demonstrate the effectiveness of our recurrent model by comparing with non-recurrent variants in Table 9. In particular, we first compare with non-weight-sharing multi-step stacked networks where different iterations have different model weights and all the other components are the same. We can observe that non-weight-sharing not only leads to $4\times$ more parameters, but also results in worse view synthesis results. We further compare with non-weight-sharing single-step deeper networks by increasing the number of attention blocks for a single-step refinement, where the results are clearly worse than our multi-step recurrent network. The weight-sharing design in our recurrent network implicitly regularizes training, which is not only more parameter-efficient but also leads to better results.

**Comparison with SplatFormer.** We note that our ReSplat has several crucial differences with SplatFormer (Chen et al., 2025a). First, we identify the rendering error as an informative feedback signal for improving the Gaussian splats, which is missing in SplatFormer. Second, ReSplat is a recurrent model which supports multi-step refinement with a weight-sharing architecture, while SplatFormer is a non-recurrent network designed for single-step refinement. Third, ReSplat is a pure feed-forward model with feed-forward initialization and feed-forward refinement. In contrast, SplatFormer relies on lengthy optimization-based 3DGS to get initial Gaussians. Fourth, SplatFormer is designed for object-centric datasets, while ReSplat can handle diverse scene-level datasets where the complexity is much higher than objects. Despite these differences, we tried to conduct a comparison with SplatFormer on our scene-level datasets. We found it particularly challenging to make it work properly for scene-level datasets, since it replies on Point Transformer V3 (Wu et al., 2024) where a proper grid size is required to serialize the point cloud. This can be done for object-centric datasets where normalizing the objects to $[-1, 1]$ is possible. However, for unbounded scene-level datasets, this would be very challenging. We tried different normalizations and did grid search for the grid size, and the best results we obtained with SplatFormer are reported in Table 10. We can see that SplatFormer is 2dB PSNR worse than our method.

**Comparison with Sparse View Optimization Methods.** We additionally compare with optimization-based 3DGS methods that are specifically designed for sparse input views. These sparse-view optimization methods (Li et al., 2024; Chung et al., 2024) usually rely on additional depth losses to regularize the optimization process. To compare with them, we perform 3DGS optimization with an additional depth loss between the rendered depth map and the estimated monocular depth map from Depth Anything V2 (Large) (Yang et al., 2024). The results are reported in Table 11. With the additional depth loss, the 3DGS optimization results are improved by 1dB PSNR. However, the gap with our ReSplat is still significant (3dB PSNR). The additionally introduced depth loss also makes the optimization slower due to the additional time for depth rendering and monocular depth estimation. In contrast, our model doesn't rely on any additional supervision from an external monocular depth model and it's $94\times$ faster thanks to our feed-forward nature.

**Features for Computing the Rendering Error.** In Table 12, we compare ResNet features with those from DINOv2. We observed no improvement when using the larger, more recent feature extractor. We attribute this to the patch-based architecture of DINOv2, which may result in coarser spatial information. In contrast, convolutional networks maintain local structural fidelity, which is critical for high-quality pixel-accurate view synthesis.

**Model Profiling.** In Table 13, we report the total runtime and individual component latency. In the initial reconstruction model, the depth prediction module constitutes the majority of the runtime. For

Table 7: **Comparison with optimization-based 3DGS across 8, 16, and 32 input views.** The quality gap becomes smaller when given 32 views for optimization-based 3DGS. However, our ReSplat is more than $200\times$ faster in terms of the reconstruction speed. The image resolution is $256 \times 448$.

| #Views | Method | Category | #Iterations | PSNR ↑ | SSIM ↑ | LPIPS ↓ | #Gaussians | Recon. Time (s) |
|---|---|---|---|---|---|---|---|---|
| 8 | 3DGS | Optimization | 4000 | 26.44 | 0.841 | 0.134 | 250K | 49 |
|  | ReSplat | Feed-Forward | **4** | **29.20** | **0.904** | **0.104** | **57K** | **0.21** |
| 16 | 3DGS | Optimization | 4000 | 27.38 | 0.864 | 0.119 | 395K | 70 |
|  | ReSplat | Feed-Forward | **4** | **29.01** | **0.900** | **0.105** | **114K** | **0.34** |
| 32 | 3DGS | Optimization | 4000 | 27.86 | 0.879 | **0.113** | 522K | 160 |
|  | ReSplat | Feed-Forward | **4** | **28.30** | **0.891** | 0.114 | **229K** | **0.75** |

Table 8: **Optimization-based *vs*. feed-forward refinement.** Starting from the same ReSplat's initialization, we compare 3DGS optimization-based refinement with our feed-forward approach. Our ReSplat improves the rendering quality significantly faster (4 *vs*. 80 iterations) and is $53\times$ faster in terms of the reconstruction speed. See Fig. 4 (main paper) for the visualization of this table.

| Method | # Iter. | PSNR ↑ | SSIM ↑ | LPIPS ↓ | Time (s) ↓ |
|---|---|---|---|---|---|
| ReSplat (init) | 0 | 26.21 | 0.842 | 0.185 | 0.311 |
| ReSplat (init) + 3DGS | 1 | 26.29 | 0.844 | 0.185 | 0.852 |
| ReSplat (init) + 3DGS | 2 | 26.44 | 0.846 | 0.183 | 1.393 |
| ReSplat (init) + 3DGS | 3 | 26.55 | 0.848 | 0.183 | 1.934 |
| ReSplat (init) + 3DGS | 4 | 26.64 | 0.849 | 0.183 | 2.475 |
| ReSplat (init) + 3DGS | 10 | 26.98 | 0.855 | 0.183 | 5.721 |
| ReSplat (init) + 3DGS | 20 | 27.29 | 0.960 | 0.183 | 11.131 |
| ReSplat (init) + 3DGS | 40 | 27.56 | 0.863 | 0.181 | 21.951 |
| ReSplat (init) + 3DGS | 60 | 27.68 | 0.865 | 0.180 | 32.771 |
| ReSplat (init) + 3DGS | 80 | **27.73** | 0.865 | 0.179 | 43.591 |
| ReSplat | 1 | 27.15 | 0.859 | 0.169 | 0.439 |
| ReSplat | 2 | 27.51 | 0.865 | 0.163 | 0.567 |
| ReSplat | 3 | 27.65 | 0.867 | 0.161 | 0.795 |
| ReSplat | 4 | 27.70 | **0.868** | **0.160** | **0.823** |

the recurrent model, the kNN attention mechanism consumes the most time. These results highlight potential areas for future optimization.

# B  ADDITIONAL DETAILS

**Training Details.** We train our model with cosine learning rate schedule. For experiments on DL3DV, we adopt a progressive training strategy with gradually increased image resolutions and number of input views for better efficiency. More specifically, we first train our model with 8 input views at $256 \times 448$ resolution, and then we fine-tune the model with 8 input views at $512 \times 960$ resolution, and finally we fine-tune the model with 16 input views at $512 \times 960$ resolution. For each stage, we train with 16 GH200 GPUs for 80K steps, with 50K steps for the initial reconstruction model and 30K steps for the recurrent model. For experiments on RealEstate10K at $256 \times 256$ resolution, we first train the initial model with 16 GH200 GPUs for 200K steps and then train the recurrent model for 100K steps.

# C  ADDITIONAL VISUALIZATIONS

We invite readers to our supplementary video to view our video rendering results.

Table 9: **Comparison of recurrent and non-recurrent architectures.** Compared to non-weight-sharing multi-step stacked networks and non-weight-sharing single-step deeper networks, our weight-sharing multi-step recurrent network is not only more parameter-efficient, but also leads to better results. We also note that our single recurrent network can support different numbers of iterations thanks to weight-sharing.

| Configuration | #Params | PSNR ↑ | SSIM ↑ | LPIPS ↓ |
|---|---|---|---|---|
| *weight-sharing* | | | | |
| Recurrent (iter 1, block 4) | 13.8M | 28.17 | 0.890 | 0.118 |
| Recurrent (iter 2, block 4) | 13.8M | 28.73 | 0.898 | 0.110 |
| Recurrent (iter 3, block 4) | 13.8M | 28.96 | 0.901 | 0.107 |
| Recurrent (iter 4, block 4) | 13.8M | **29.07** | **0.902** | **0.105** |
| *non-weight-sharing, multi-step, stacked* | | | | |
| Non-recurrent (stack 1) | 13.8M | 28.17 | 0.890 | 0.118 |
| Non-recurrent (stack 2) | 27.6M | 28.74 | 0.898 | 0.109 |
| Non-recurrent (stack 3) | 41.4M | 28.72 | 0.898 | 0.110 |
| Non-recurrent (stack 4) | 55.2M | 28.71 | 0.897 | 0.110 |
| *non-weight-sharing, single-step, deeper* | | | | |
| Non-recurrent (stack 1, block 4) | 13.8M | 28.17 | 0.890 | 0.118 |
| Non-recurrent (stack 1, block 8) | 27.6M | 28.30 | 0.891 | 0.116 |
| Non-recurrent (stack 1, block 12) | 41.4M | 28.36 | 0.893 | 0.115 |
| Non-recurrent (stack 1, block 16) | 55.2M | 28.40 | 0.893 | 0.115 |

Table 10: **Comparison with SplatFormer.** It's non-trivial to make SplatFormer work properly for scene-level datasets, while our ReSplat is 2dB PSNR better.

| Method | PSNR ↑ | SSIM ↑ | LPIPS ↓ |
|---|---|---|---|
| SplatFormer | 27.03 | 0.868 | 0.140 |
| ReSplat | **29.07** | **0.902** | **0.105** |

In Fig. 8, we show more visual comparisons with 3DGS, MVSplat and DepthSplat on DL3DV dataset.

In Fig. 9, we show the visual results with different numbers of iterations.

In Fig. 10, we show the visual ablation results of our initial model. In Fig. 11, we show the visual ablation results of our recurrent model.

In Fig. 7, we show the visual results of different compression factors, and the detailed metrics are in Table 5 (main paper). We choose $16\times$ compression since it offers a good trade-off between speed and accuracy.

We provide visualizations of reconstructed Gaussians in Fig. 12 and Fig. 13, where we can observe that our ReSplat reconstructs cleaner Gaussians and renders higher-quality images than DepthSplat.

## D    USE OF LARGE LANGUAGE MODELS (LLMS)

We used LLMs only for polishing text (grammar, readability, and style) and for assistance with figure creation and LaTeX formatting. The models were not used for idea generation, technical content, or experimental design. All scientific contributions and results in this paper are solely from the authors.

Table 11: **Comparison with depth-regularized 3DGS optimization method.** Despite with an additional depth loss, the optimization-based method is still 3dB PSNR worse than our ReSplat and runs 94× slower.

| Method | PSNR ↑ | SSIM ↑ | LPIPS ↓ | Recon. Time (s) |
|---|---|---|---|---|
| 3DGS (w/o depth loss) | 23.46 | 0.770 | 0.224 | 70.0 |
| 3DGS (w/ depth loss) | 24.54 | 0.796 | 0.204 | 75.4 |
| ReSplat (w/o depth loss) | **27.70** | **0.868** | **0.160** | **0.8** |

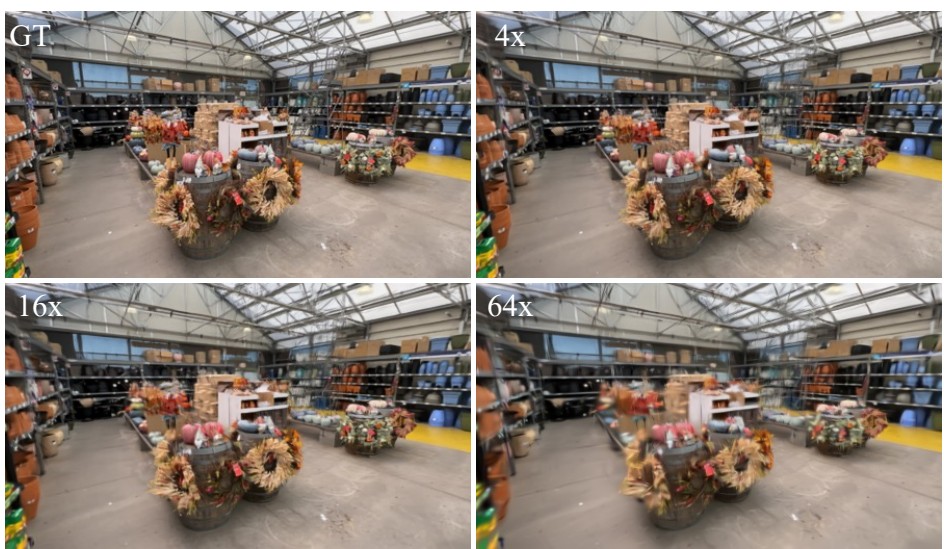

Figure 7: **Different compression factors.** 16× compression represents a good speed-accuracy trade-off. See Table 5 for detailed metrics.

Table 12: **ResNet vs. DINOv2 features for computing the rendering error.** We observed no improvement when using the larger, more recent feature extractor. We attribute this to the patch-based architecture of DINOv2, which may result in coarser spatial information. In contrast, convolutional networks maintain local structural fidelity, which is critical for high-quality view synthesis.

| Features | #Parameters | PSNR ↑ | SSIM ↑ | LPIPS ↓ |
|---|---|---|---|---|
| ResNet | 0.7M | 29.07 | 0.902 | 0.105 |
| DINOv2 | 86.6M | 29.00 | 0.901 | 0.107 |

Table 13: **Model Profiling.** Inference time (s) measured on 8 input views at varying resolutions. We report both total runtime and individual component latency.

(a) **Initial Model Profiling.** The depth prediction module constitutes the majority of the runtime.

| Resolution | Total | Depth pred. | kNN attn | Global attn | Gaussian head |
|---|---|---|---|---|---|
| 256 × 448 | 0.149 | 0.111 | 0.024 | 0.013 | 0.001 |
| 512 × 960 | 0.311 | 0.197 | 0.094 | 0.018 | 0.002 |

(b) **Recurrent Model Profiling.** The kNN attention mechanism consumes the most time.

| Resolution | Total | Render error | kNN attn | Global attn | Update head |
|---|---|---|---|---|---|
| 256 × 448 | 0.022 | 0.003 | 0.015 | 0.002 | 0.002 |
| 512 × 960 | 0.126 | 0.016 | 0.092 | 0.008 | 0.010 |

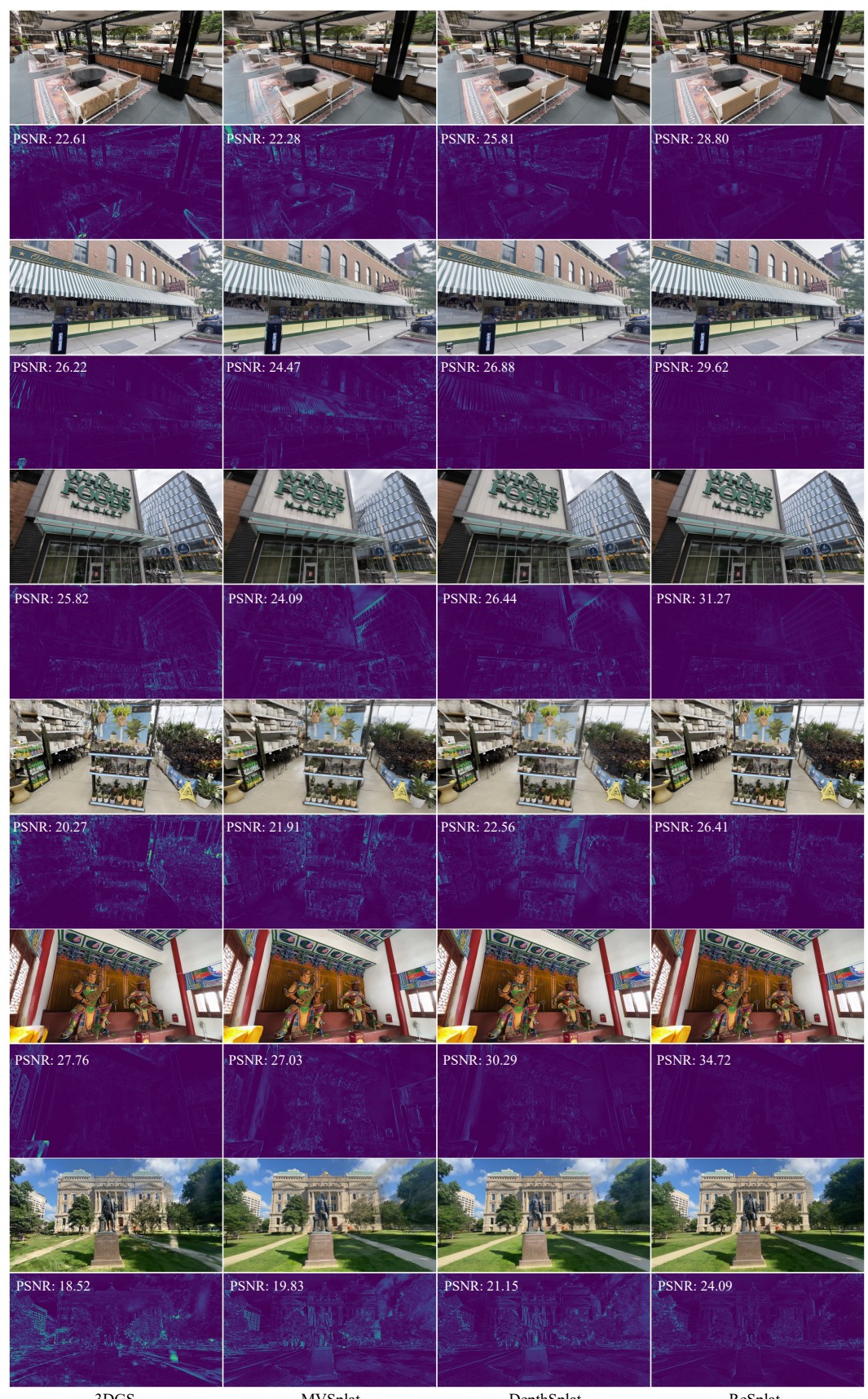

Figure 8: **More comparisons of novel view synthesis on DL3DV**.

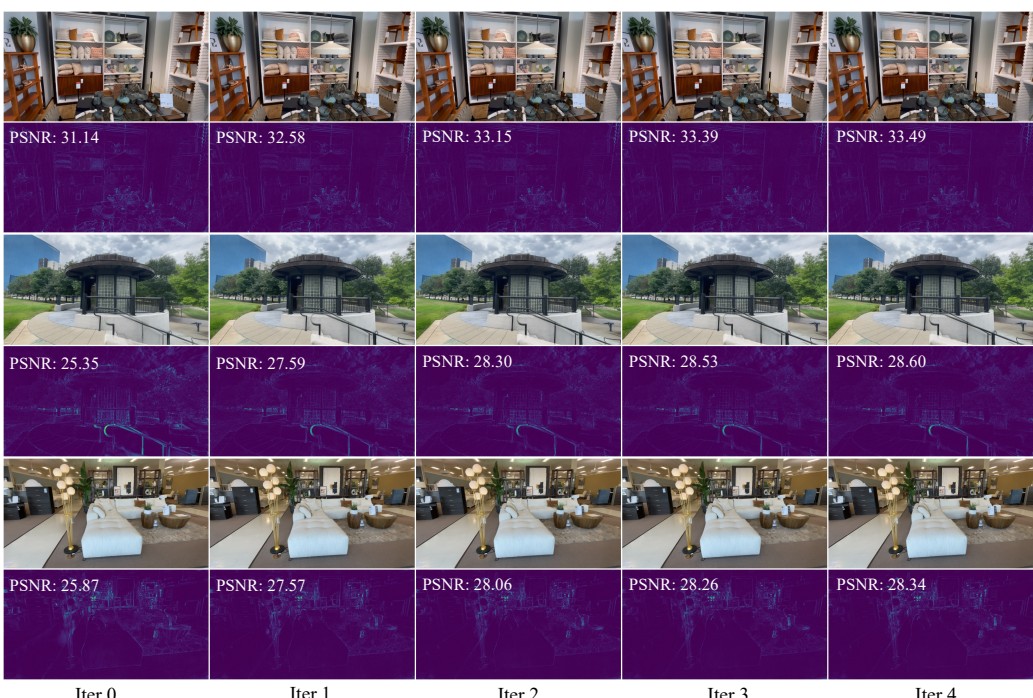

Figure 9: **Results with different numbers of iterations**.

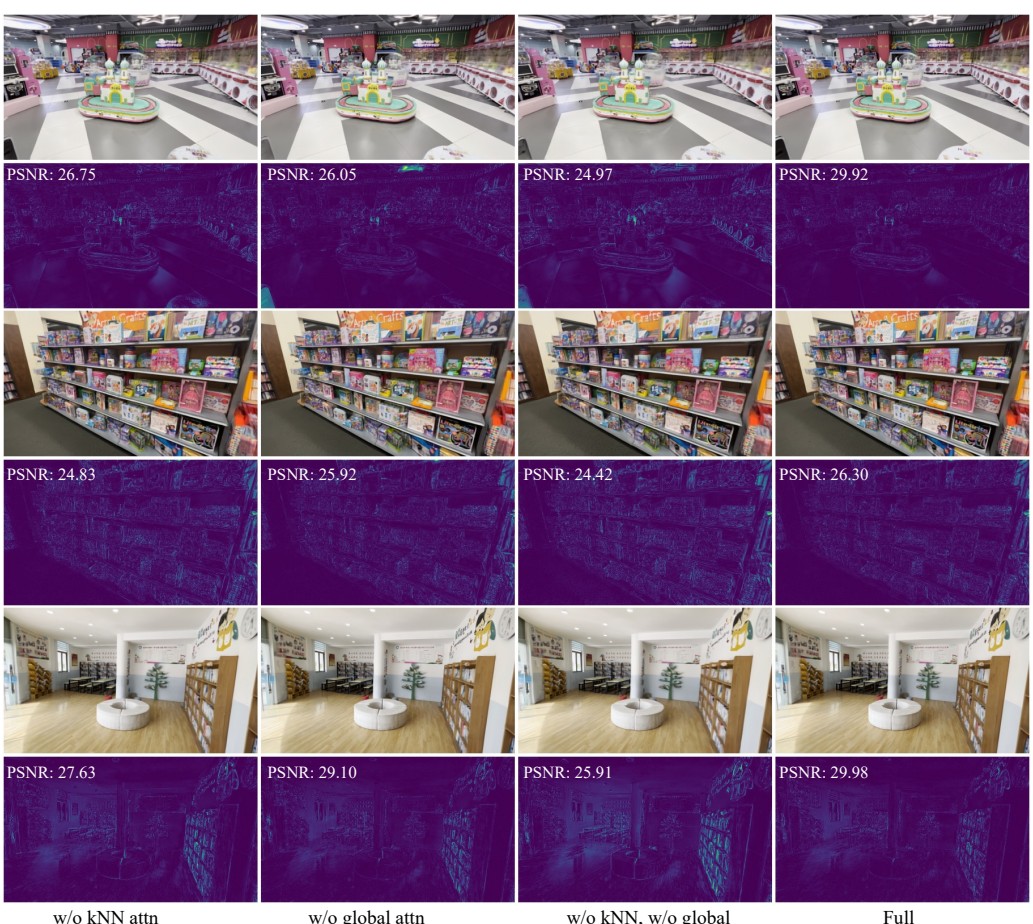

Figure 10: **Ablation of the initial model**.

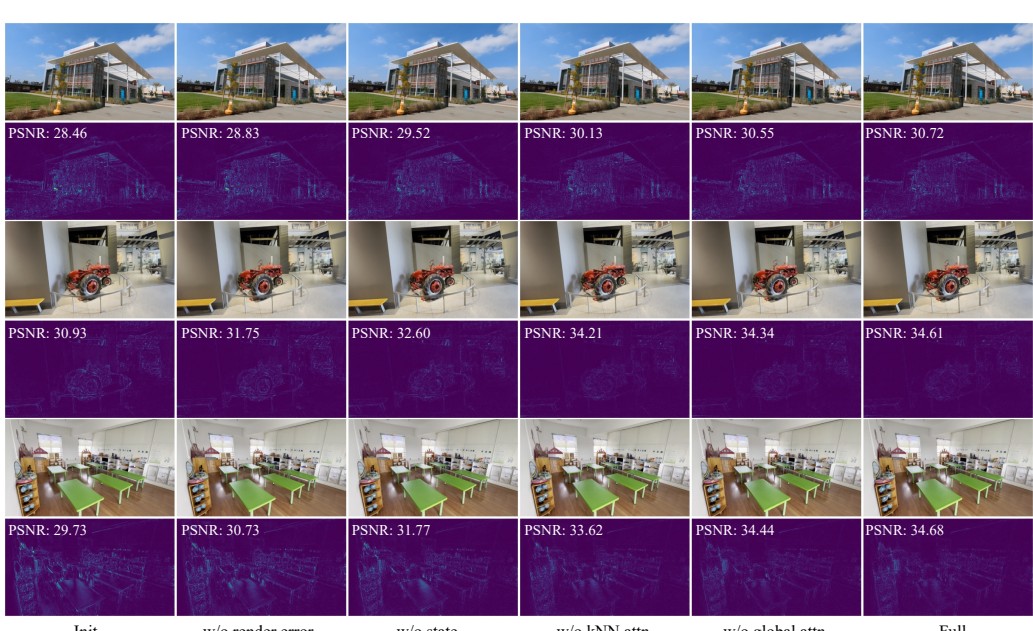

Figure 11: **Ablation of the recurrent model**.

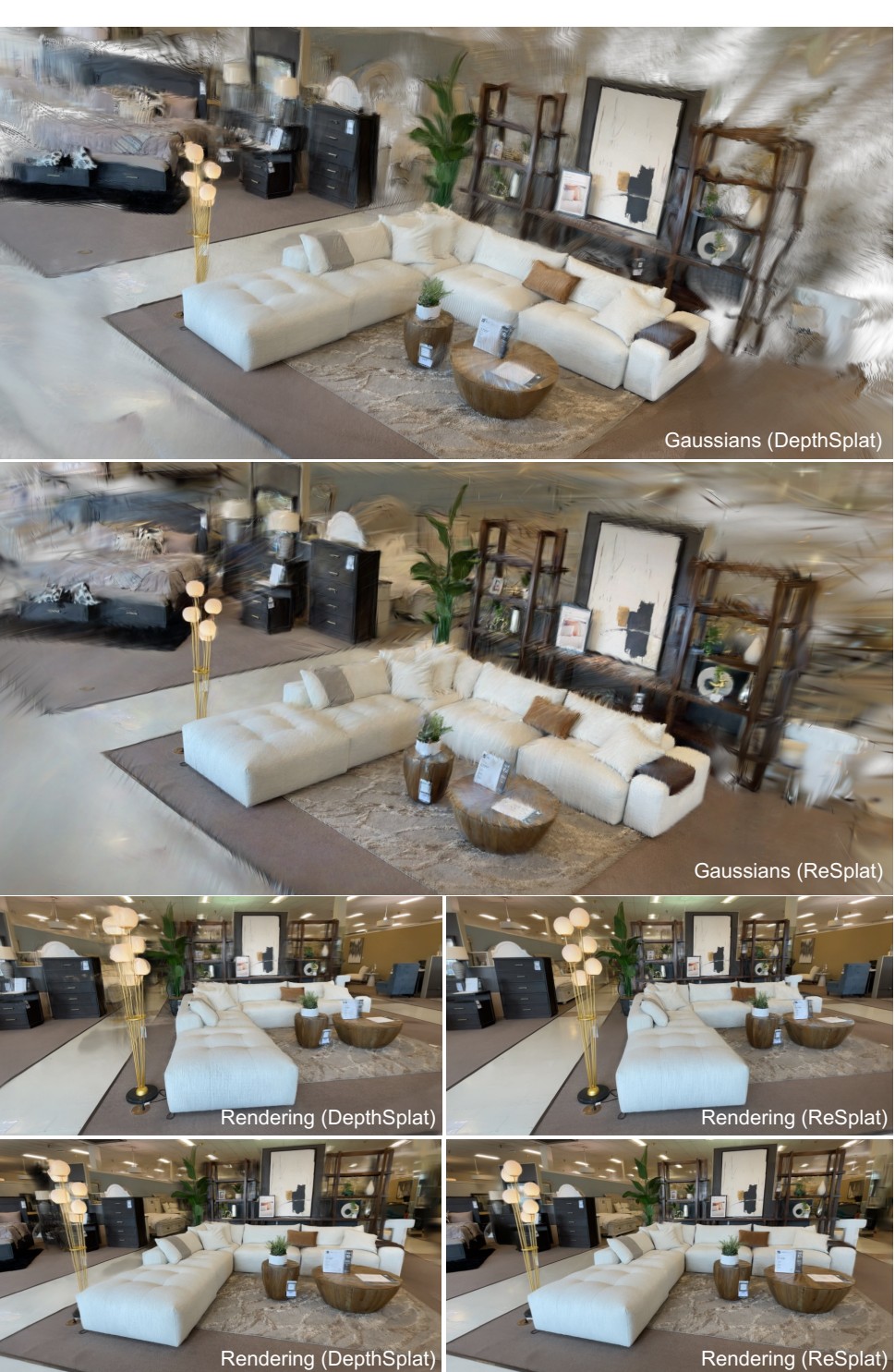

Figure 12: **Comparison of reconstructed Gaussians and view synthesis results**.

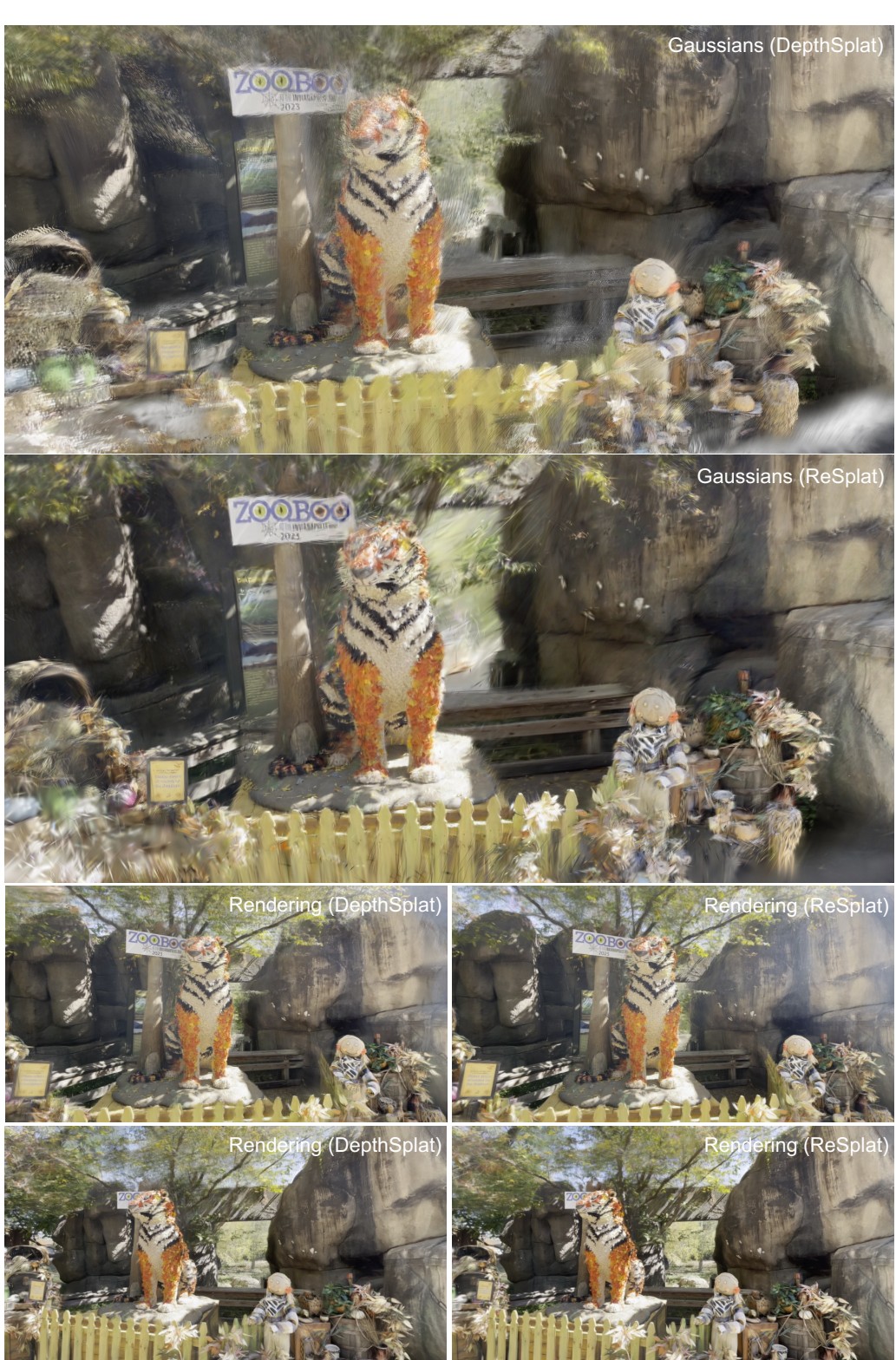

Figure 13: **Comparison of reconstructed Gaussians and view synthesis results**.

