# OpenReview forum: "ReSplat: Learning Recurrent Gaussian Splats"
_ICLR.cc/2026/Conference — Submitted to ICLR 2026_

### Official Review · Reviewer_ZHGA · 2025-10-25

**Soundness:** 2
**Presentation:** 3
**Contribution:** 2
**Rating:** 4
**Confidence:** 4

**Summary:**

The paper proposes a feed-forward reconstruction paradigm based on 3DGS, whose core contributions include a single-step feed-forward Gaussian reconstruction model with compression capabilities to initialize the recurrent process, as well as a weight-sharing recurrent network that iteratively improves the reconstruction. The authors claim that their method achieves better performance compared to existing works.

**Strengths:**

1. This work attempts to address the scalability issues caused by per-pixel Gaussian to effectively reduce the number of Gaussians, which is an important problem for feed-forward reconstruction.

2. The proposed initialization model is effective, achieving both performance improvement and compression of the number of Gaussians.

3. The paper provides extensive quantitative experiments that validate the effectiveness of the method.

**Weaknesses:**

1. The qualitative experiments presented in the main text and appendix are limited, especially the qualitative comparison experiments under different iteration numbers, such as Figure 1. The limited cases are insufficient to demonstrate the effectiveness of this iterative refinement. Furthermore, qualitative ablation experiments for the different proposed components are missing, making it difficult to intuitively analyze what role these modules play.

2. The core contribution of the paper lies in proposing a compressible initial feed-forward reconstruction and a weight-sharing recurrent network, where the latter is similar to SplatFormer and LIFe-GOM. Although the authors explain that the application scenarios differ, this makes me question whether the contribution of this work is limited. Furthermore, the experimental section lacks comparisons with these similar methods (e.g., Initialization + SplatFormer), which affects the evaluation of the proposed method's effectiveness.

3. The working mechanism underlying the proposed iterative refinement module remains unclear. While its overall framework resembles the initialization stage, it achieves substantial performance improvements at inference time without introducing any additional auxiliary information or optimization. I suspect that this improvement may stem from stacking Attention blocks, similar to scaling laws. If this is the case, the contribution would be trivial.

4. Although the authors discuss cross-dataset generalization in Figure 4, it lacks comparative experiments on cross-dataset scenarios, such as ACID, which has been mentioned in many existing works.

5. Table 1 and Table 2 discuss comparisons with some optimization-based methods, but the methods mentioned are based on dense views, such as 3DGS and Mip-Splatting. Since the experimental setting is based on sparse views (8 or 16 views), such an experimental setup is unfair. It is recommended to introduce comparisons with some optimization-based sparse-view reconstruction works.

6. Compared to existing methods like MVSplat and DepthSplat, ReSplat's efficiency is negatively affected, which may stem from the introduction of kNN attention.

**Questions:**

1. In the recurrent reconstruction model, I observe that the impact of global attention is limited. Is this module necessary? Does its structure need to remain consistent with the initialization model?

2. Compared to existing algorithms, what is ReSplat's inference computational cost?

3. Why does ReSplat compress the number of Gaussians by 16×, yet only leads to 4× faster rendering speed?

---

> ### Author Response · Authors · 2025-12-03
> **Official Response by Authors (1/4)**
>
> > The qualitative experiments presented in the main text and appendix are limited, especially the qualitative comparison experiments under different iteration numbers, such as Figure 1. The limited cases are insufficient to demonstrate the effectiveness of this iterative refinement. Furthermore, qualitative ablation experiments for the different proposed components are missing, making it difficult to intuitively analyze what role these modules play.
>
> We thank the reviewer for the suggestion and we have included more qualitative results in Figure 8, Figure 9, Figure 10, and Figure 11 of the revised manuscript.
>
> > The core contribution of the paper lies in proposing a compressible initial feed-forward reconstruction and a weight-sharing recurrent network, where the latter is similar to SplatFormer and LIFe-GOM. Although the authors explain that the application scenarios differ, this makes me question whether the contribution of this work is limited. Furthermore, the experimental section lacks comparisons with these similar methods (e.g., Initialization + SplatFormer), which affects the evaluation of the proposed method's effectiveness.
>
> We note that our ReSplat has several crucial differences with SplatFormer:
>
> - First, we identify the rendering error as an informative feedback signal for improving the Gaussian splats, which is missing in SplatFormer.
> - Second, ReSplat is a recurrent model which supports multi-step refinement with a weight-sharing architecture, while SplatFormer is a non-recurrent network designed for single-step refinement.
> - Third, ReSplat is a pure feed-forward model with feed-forward initialization and feed-forward refinement. In contrast, SplatFormer relies on lengthy optimization-based 3DGS to get initial Gaussians.
> - Fourth, SplatFormer is designed for object-centric datasets, while ReSplat can handle diverse scene-level datasets where the complexity is much higher than objects.
>
> Despite these differences, we tried to conduct a comparison with SplatFormer on our scene-level datasets. We found it particularly challenging to make it work properly for scene-level datasets, since it replies on Point Transformer V3 where a proper grid size is required to serialize the point cloud. This can be done for object-centric datasets where normalizing the objects to $[-1, 1]$ is possible. However, for unbounded scene-level datasets, this would be very challenging. We tried different normalizations and did grid search for the grid size, and the best results we obtained with SplatFormer are reported below. We can see that SplatFormer is still 2dB PSNR worse than our method.
>
> | Method      |  PSNR ↑   |  SSIM ↑   |  LPIPS ↓  |
> | :---------- | :-------: | :-------: | :-------: |
> | SplatFormer |   27.03   |   0.868   |   0.140   |
> | ReSplat     | **29.07** | **0.902** | **0.105** |
>
> These results are included in Table 10 of the revised manuscript and the discussions are added in L723-737.

---

> > ### Author Response · Authors · 2025-12-03
> > **Official Response by Authors (2/4)**
> >
> > > The working mechanism underlying the proposed iterative refinement module remains unclear. While its overall framework resembles the initialization stage, it achieves substantial performance improvements at inference time without introducing any additional auxiliary information or optimization. I suspect that this improvement may stem from stacking Attention blocks, similar to scaling laws. If this is the case, the contribution would be trivial.
> >
> > We would like to highlight that the rendering error plays an indispensable role in our model, as can be seen from the extensive ablations and analysis in Table 6 of our manuscript. The rendering error provides an informative feedback to the recurrent model, which guides the network to learn effective Gaussian updates.
> >
> > Regarding the recurrent architecture, we compare with non-recurrent variants by stacking more attention blocks. In particular, we first compare with non-weight-sharing multi-step stacked networks where different iterations have different model weights and all the other components are the same with our ReSplat. We can observe that non-weight-sharing not only leads to $4 \times$ more parameters, but also results in worse view synthesis results. We further compare with non-weight-sharing single-step deeper networks by increasing the number of attention blocks for a single-step refinement, where the results are clearly worse than our multi-step recurrent network. The weight-sharing design in our recurrent network implicitly regularizes training, which is not only more parameter-efficient but also leads to better results.
> >
> > | **Configuration**                         | **#Params** | **PSNR ↑** | **SSIM ↑** | **LPIPS ↓** |
> > | ----------------------------------------- | ----------- | ---------- | ---------- | ----------- |
> > | *weight-sharing*                          |             |            |            |             |
> > | Recurrent (iter 1, block 4)               | 13.8M       | 28.17      | 0.890      | 0.118       |
> > | Recurrent (iter 2, block 4)               | 13.8M       | 28.73      | 0.898      | 0.110       |
> > | Recurrent (iter 3, block 4)               | 13.8M       | 28.96      | 0.901      | 0.107       |
> > | Recurrent (iter 4, block 4)               | 13.8M       | **29.07**  | **0.902**  | **0.105**   |
> > | *non-weight-sharing, multi-step, stacked* |             |            |            |             |
> > | Non-recurrent (stack 1)                   | 13.8M       | 28.17      | 0.890      | 0.118       |
> > | Non-recurrent (stack 2)                   | 27.6M       | 28.74      | 0.898      | 0.109       |
> > | Non-recurrent (stack 3)                   | 41.4M       | 28.72      | 0.898      | 0.110       |
> > | Non-recurrent (stack 4)                   | 55.2M       | 28.71      | 0.897      | 0.110       |
> > | *non-weight-sharing, single-step, deeper* |             |            |            |             |
> > | Non-recurrent (stack 1, block 4)          | 13.8M       | 28.17      | 0.890      | 0.118       |
> > | Non-recurrent (stack 1, block 8)          | 27.6M       | 28.30      | 0.891      | 0.116       |
> > | Non-recurrent (stack 1, block 12)         | 41.4M       | 28.36      | 0.893      | 0.115       |
> > | Non-recurrent (stack 1, block 16)         | 55.2M       | 28.40      | 0.893      | 0.115       |
> >
> > These results demonstrate the effectiveness of our recurrent model and they are included in Table 9 of our revised manuscript and the discussions are added in L714-722.

---

> > > ### Author Response · Authors · 2025-12-03
> > > **Official Response by Authors (3/4)**
> > >
> > > > Although the authors discuss cross-dataset generalization in Figure 4, it lacks comparative experiments on cross-dataset scenarios, such as ACID, which has been mentioned in many existing works.
> > >
> > > We show the zero-shot cross-dataset generalization results below, where our ReSplat outperforms previous methods by 1dB PSNR, demonstrating the generalization of our method.
> > >
> > > These results are included in Table 4 of the revised manuscript.
> > >
> > > | Method     |  PSNR ↑   |  SSIM ↑   |  LPIPS ↓  |
> > > | :--------- | :-------: | :-------: | :-------: |
> > > | pixelSplat |   27.64   |   0.830   |   0.160   |
> > > | MVSplat    |   28.15   |   0.841   |   0.147   |
> > > | DepthSplat |   28.37   |   0.847   |   0.141   |
> > > | GS-LRM     |   28.84   |   0.849   |   0.146   |
> > > | ReSplat    | **29.87** | **0.864** | **0.135** |
> > >
> > > > Table 1 and Table 2 discuss comparisons with some optimization-based methods, but the methods mentioned are based on dense views, such as 3DGS and Mip-Splatting. Since the experimental setting is based on sparse views (8 or 16 views), such an experimental setup is unfair. It is recommended to introduce comparisons with some optimization-based sparse-view reconstruction works.
> > >
> > > We additionally compare with optimization-based 3DGS methods that are specifically designed for sparse input views. These sparse-view optimization methods usually rely on additional depth losses to regularize the optimization process. To compare with them, we perform 3DGS optimization with an additional depth loss between the rendered depth map and the estimated monocular depth map from Depth Anything V2 (Large). The results are reported below.
> > >
> > > | **Method**               | **PSNR ↑** | **SSIM ↑** | **LPIPS ↓** | **Recon. Time (s)** |
> > > | ------------------------ | ---------- | ---------- | ----------- | ------------------- |
> > > | 3DGS (w/o depth loss)    | 23.46      | 0.770      | 0.224       | 70.0                |
> > > | 3DGS (w/ depth loss)     | 24.54      | 0.796      | 0.204       | 75.4                |
> > > | ReSplat (w/o depth loss) | **27.70**  | **0.868**  | **0.160**   | **0.8**             |
> > >
> > > With the additional depth loss, the 3DGS optimization results are improved by 1dB PSNR. However, the gap with our ReSplat is still significant (3dB PSNR). The additionally introduced depth loss also makes the optimization slower due to the additional time for depth rendering and monocular depth estimation. In contrast, our model doesn't rely on any additional supervision from an external monocular depth model and it's $94\times$ faster thanks to our feed-forward nature.
> > >
> > > There results are included in Table 11 of our revised manuscript and the discussions are added in L739-748.

---

> > > > ### Author Response · Authors · 2025-12-03
> > > > **Official Response by Authors (4/4)**
> > > >
> > > > > Compared to existing methods like MVSplat and DepthSplat, ReSplat's efficiency is negatively affected, which may stem from the introduction of kNN attention.
> > > >
> > > > Compared to MVSplat and DepthSplat, ReSplat derives its efficiency from a $16\times$ reduction in Gaussian parameters and $4\times$ faster rendering speeds, enabling superior scalability across many input views and high-resolution images. Although our reconstruction requires more time, the resulting Gaussian output is significantly more compact, which is a crucial factor for supporting interactive applications. Furthermore, we achieve substantial quality improvements, outperforming MVSplat by +5.2dB PSNR and DepthSplat by +3.4dB PSNR. Finally, our recurrent model demonstrates robust generalization across unseen datasets, view counts and image resolutions, overcoming the limitations faced by previous methods like MVSplat and DepthSplat.
> > > >
> > > > > In the recurrent reconstruction model, I observe that the impact of global attention is limited. Is this module necessary? Does its structure need to remain consistent with the initialization model?
> > > >
> > > > The motivation of using global attention is to propagate the rendering error globally to the 3D Gaussians. With simple concatenation, the information received by Gaussians is local and per-view. The multi-view global attention provides additional global context which is complementary to the per-view local ResNet features. Our global attention is lightweight, since it's just a single global attention layer implemented with Flash Attention 3. From Figure 11 of the visual results in our manuscript, we can observe that the global attention improves the global structure coherence, and local kNN attention contributes to the local details. The structure of the recurrent network is independent with the initialization model, and they are trained separately.
> > > >
> > > > > Compared to existing algorithms, what is ReSplat's inference computational cost?
> > > >
> > > > Compared to MVSplat and DepthSplat, our reconstruction time is $2-4\times$ slower but ReSplat produces $16\times$ fewer Gaussians and enables $4\times$ faster rendering speed. Compared to optimization-based 3DGS, our reconstruction time is $100\times$ faster.
> > > >
> > > > The detailed numbers are reported in Table 1, Table 2, Figure 4(b) of our manuscript.
> > > >
> > > > > Why does ReSplat compress the number of Gaussians by 16×, yet only leads to 4× faster rendering speed?
> > > >
> > > > While rendering speed is related to the total number of Gaussians, the relationship is not linear. The performance is also influenced by the screen-space size and distribution of the Gaussians. Specifically, ReSplat's average Gaussian scale is $3.95\times$ larger than that of MVSplat and DepthSplat. These larger primitives are necessary to maintain coverage with fewer Gaussians. Consequently, this increases the number of Gaussians to traverse and alpha-blend for each pixel, which offsets the performance gains of rendering speed from the reduced total number of Gaussians.

---

### Official Review · Reviewer_45AN · 2025-10-31

**Soundness:** 2
**Presentation:** 3
**Contribution:** 2
**Rating:** 4
**Confidence:** 4

**Summary:**

This paper proposes a method that recurrently refines Gaussians generated by feed-forward models using rendering error as a feedback signal. The initial Gaussians are produced in a 16× subsampled space to reduce computational cost during refinement. The proposed approach improves reconstruction quality on both in-domain and cross-domain datasets.

**Strengths:**

- This paper is well written, with clear motivation and a well-reasoned design.
- The experiments are comprehensive, demonstrating not only the effectiveness of the proposed method but also its generalizability to unseen data and unseen resolution.

**Weaknesses:**

- Although the experimental results show that reconstruction quality improves with more refinement iterations of the initial Gaussians, a similar effect could also be achieved with the original optimization-based refinement. Traditional optimization indeed converges slowly due to the sparse SfM initialization, but recent works such as Long-LRM have shown that initialization using dense Gaussian outputs from a feed-forward model can significantly reduce optimization steps. A comparison with such a scenario would make the proposed method’s advantage more convincing.
- The two main contributions of the paper are the Gaussian initialization strategy and the recurrent network. However, the performance gain seems largely attributed to the point-level architecture and the use of rendering error for refinement, rather than to the recurrent nature itself. As shown in Table 1, the initial model improves over DepthSplat by more than 2 dB in PSNR, primarily due to these architectural enhancements, which is further supported by the ablation in Table 5. While the recurrent network provides additional gains (Tables 1, 2, and 5), the improvements saturate after two or three iterations, as noted in the limitations section. This raises the question of whether the recurrent design is essential. It would strengthen the paper to provide an explanation or intuition for why the improvement plateaus after a few iterations.
- Conceptually, the proposed method appears to serve as an efficient refiner of Gaussians initialized by a feed-forward model—an alternative to lengthy gradient-based optimization. In this regard, it would be valuable if the method could be applied directly to existing feed-forward models without retraining. While I believe the proposed refining network can be integrated into the original-resolution feed-forward models, this could potentially increase memory usage and computational cost, which may affect the method’s praticaility.

**Questions:**

- The recurrent network takes a hidden state as input. From my understanding, the Gaussian parameters themselves evolve over iterations and could naturally serve as the hidden state, since they are updated and reused. Could the authors clarify why a separate hidden state is necessary and how it is initialized?
- The paper reports strong generalization to unseen resolutions. I understand that the recurrent naturally allow adaptation to unseen data distributions (e.g., RealEstate10K) by iteratively refining Gaussians using rendering error, but it is unclear how this mechanism supports resolution generalization. Could the authors elaborate on this aspect?

---

> ### Author Response · Authors · 2025-12-03
> **Official Response by Authors (1/4)**
>
> >  Although the experimental results show that reconstruction quality improves with more refinement iterations of the initial Gaussians, a similar effect could also be achieved with the original optimization-based refinement. Traditional optimization indeed converges slowly due to the sparse SfM initialization, but recent works such as Long-LRM have shown that initialization using dense Gaussian outputs from a feed-forward model can significantly reduce optimization steps. A comparison with such a scenario would make the proposed method’s advantage more convincing.
>
> In Figure 4 and Table 8 of the revised manuscript, we compare optimization-based refinement with our feed-forward refinement by using the same ReSplat's feed-forward initialization. Our ReSplat improves the rendering quality significantly faster than optimization (4 vs. 80 iterations) and is $53\times $ faster in terms of the reconstruction speed. These results demonstrate the high efficiency and quality of our feed-forward refinement method.
>
> |                       | #Iterations |   PSNR    |   SSIM    |   LPIPS   | Time (s)  |
> | :-------------------: | :---------: | :-------: | :-------: | :-------: | :-------: |
> |    ReSplat (init)     |      0      |   26.21   |   0.842   |   0.185   |   0.311   |
> |                       |             |           |           |           |           |
> | ReSplat (init) + 3DGS |      1      |   26.29   |   0.844   |   0.185   |   0.852   |
> | ReSplat (init) + 3DGS |      2      |   26.44   |   0.846   |   0.183   |   1.393   |
> | ReSplat (init) + 3DGS |      3      |   26.55   |   0.848   |   0.183   |   1.934   |
> | ReSplat (init) + 3DGS |      4      |   26.64   |   0.849   |   0.183   |   2.475   |
> | ReSplat (init) + 3DGS |     10      |   26.98   |   0.855   |   0.183   |   5.721   |
> | ReSplat (init) + 3DGS |     20      |   27.29   |   0.960   |   0.183   |  11.131   |
> | ReSplat (init) + 3DGS |     40      |   27.56   |   0.863   |   0.181   |  21.951   |
> | ReSplat (init) + 3DGS |     60      |   27.68   |   0.865   |   0.180   |  32.771   |
> | ReSplat (init) + 3DGS |     80      | **27.73** |   0.865   |   0.179   |  43.591   |
> |                       |             |           |           |           |           |
> |        ReSplat        |      1      |   27.15   |   0.859   |   0.169   |   0.439   |
> |        ReSplat        |      2      |   27.51   |   0.865   |   0.163   |   0.567   |
> |        ReSplat        |      3      |   27.65   |   0.867   |   0.161   |   0.795   |
> |        ReSplat        |      **4**      |   27.70   | **0.868** | **0.160** | **0.823** |

---

> > ### Author Response · Authors · 2025-12-03
> > **Official Response by Authors (2/4)**
> >
> > > The two main contributions of the paper are the Gaussian initialization strategy and the recurrent network. However, the performance gain seems largely attributed to the point-level architecture and the use of rendering error for refinement, rather than to the recurrent nature itself. As shown in Table 1, the initial model improves over DepthSplat by more than 2 dB in PSNR, primarily due to these architectural enhancements, which is further supported by the ablation in Table 5. While the recurrent network provides additional gains (Tables 1, 2, and 5), the improvements saturate after two or three iterations, as noted in the limitations section. This raises the question of whether the recurrent design is essential. It would strengthen the paper to provide an explanation or intuition for why the improvement plateaus after a few iterations.
> >
> > We thank the reviewer for the suggestion. We did additional analysis and found out the choice of a global coordinate system matters for our recurrent network. In particular, since our recurrent network operates within a global 3D space, the selection of a coordinate system directly determines the spatial distribution of the Gaussian's centers. Our datasets consist of video sequences with camera poses estimated from COLMAP. We evaluated aligning the global reference frame to the first, middle, and last views of the input images. Empirically, we observed that using the middle input view as the reference coordinate system yields the best performance. We posit that this centers the coordinate system, reducing the maximum transformation distance to the first and last input views and effectively balancing the spatial positions of the 3D Gaussians.
> >
> > | Reference Coordinate | **PSNR ↑** | **SSIM ↑** | **LPIPS ↓** |
> > | -------------------- | ---------- | ---------- | ----------- |
> > | Initialization       | 26.77      | 0.865      | 0.142       |
> > | COLMAP               | 28.14      | 0.886      | 0.116       |
> > | First view           | 28.66      | 0.896      | 0.109       |
> > | Last view            | 28.59      | 0.895      | 0.110       |
> > | Middle view          | **29.07**  | **0.902**  | **0.105**   |
> >
> > By just changing the reference coordinate system to the middle view, our recurrent model is further boosted by +0.9dB PSNR.
> >
> > These results are included in Table 6(b) of the revised manuscript and the discussions are added in L309-316 and L509-512.
> >
> > We also did additional analysis on the rendering error considering its importance to our model. We found that a combination of pixel-space and feature-space rendering errors (with addition) performs best. Below are the detailed numbers:
> >
> > | **Method**                    | **PSNR ↑** | **SSIM ↑** | **LPIPS ↓** |
> > | ----------------------------- | ---------- | ---------- | ----------- |
> > | Initialization                | 26.77      | 0.865      | 0.142       |
> > | w/o rendering error           | 27.19      | 0.873      | 0.137       |
> > | RGB error only                | 27.90      | 0.882      | 0.130       |
> > | Feature error only            | 28.77      | 0.897      | 0.110       |
> > | Concat (RGB & feature errors) | 28.93      | 0.900      | 0.106       |
> > | Add (RGB & feature errors)    | **29.07**  | **0.902**  | **0.105**   |
> >
> > These results are included in Table 6(a) of revised manuscript, and the descriptions and discussions are updated in L216-238 and L504-507.

---

> > > ### Author Response · Authors · 2025-12-03
> > > **Official Response by Authors (3/4)**
> > >
> > > > While the recurrent network provides additional gains (Tables 1, 2, and 5), the improvements saturate after two or three iterations, as noted in the limitations section. This raises the question of whether the recurrent design is essential. It would strengthen the paper to provide an explanation or intuition for why the improvement plateaus after a few iterations.
> > >
> > > Regarding the effect of the recurrent architecture, we compare with non-recurrent variants. In particular, we first compare with non-weight-sharing multi-step stacked networks where different iterations have different model weights and all the other components are the same with our ReSplat. We can observe that non-weight-sharing not only leads to $4 \times$ more parameters, but also results in worse view synthesis results. We further compare with non-weight-sharing single-step deeper networks by increasing the number of attention blocks for a single-step refinement, where the results are clearly worse than our multi-step recurrent network. The weight-sharing design in our recurrent network implicitly regularizes training, which is not only more parameter-efficient but also leads to better results.
> > >
> > > | **Configuration**                         | **#Params** | **PSNR ↑** | **SSIM ↑** | **LPIPS ↓** |
> > > | ----------------------------------------- | ----------- | ---------- | ---------- | ----------- |
> > > | *weight-sharing*                          |             |            |            |             |
> > > | Recurrent (iter 1, block 4)               | 13.8M       | 28.17      | 0.890      | 0.118       |
> > > | Recurrent (iter 2, block 4)               | 13.8M       | 28.73      | 0.898      | 0.110       |
> > > | Recurrent (iter 3, block 4)               | 13.8M       | 28.96      | 0.901      | 0.107       |
> > > | Recurrent (iter 4, block 4)               | 13.8M       | **29.07**  | **0.902**  | **0.105**   |
> > > | *non-weight-sharing, multi-step, stacked* |             |            |            |             |
> > > | Non-recurrent (stack 1)                   | 13.8M       | 28.17      | 0.890      | 0.118       |
> > > | Non-recurrent (stack 2)                   | 27.6M       | 28.74      | 0.898      | 0.109       |
> > > | Non-recurrent (stack 3)                   | 41.4M       | 28.72      | 0.898      | 0.110       |
> > > | Non-recurrent (stack 4)                   | 55.2M       | 28.71      | 0.897      | 0.110       |
> > > | *non-weight-sharing, single-step, deeper* |             |            |            |             |
> > > | Non-recurrent (stack 1, block 4)          | 13.8M       | 28.17      | 0.890      | 0.118       |
> > > | Non-recurrent (stack 1, block 8)          | 27.6M       | 28.30      | 0.891      | 0.116       |
> > > | Non-recurrent (stack 1, block 12)         | 41.4M       | 28.36      | 0.893      | 0.115       |
> > > | Non-recurrent (stack 1, block 16)         | 55.2M       | 28.40      | 0.893      | 0.115       |
> > >
> > > These results verify the effectiveness of our recurrent model and they are included in Table 9 of our revised manuscript and the discussions are added in L714-722.
> > >
> > > Collectively, these results demonstrate that our recurrent model improves initialization by +2.3dB PSNR, a significant margin. This performance gain is primarily attributed to the contributions introduced in this work: the rendering error and the weight-sharing recurrent architecture.
> > >
> > > To further improve the quality of our recurrent model in the future, we believe one promising direction could be exploring more adaptive refinement method. Our current model keeps the number of Gaussians fixed during refinement, introducing additional Gaussian pruning and densification mechanisms could potentially improve the results further.

---

> > > > ### Author Response · Authors · 2025-12-03
> > > > **Official Response by Authors (4/4)**
> > > >
> > > > > Conceptually, the proposed method appears to serve as an efficient refiner of Gaussians initialized by a feed-forward model—an alternative to lengthy gradient-based optimization. In this regard, it would be valuable if the method could be applied directly to existing feed-forward models without retraining. While I believe the proposed refining network can be integrated into the original-resolution feed-forward models, this could potentially increase memory usage and computational cost, which may affect the method’s praticaility.
> > > >
> > > > We would like to note that pixel-aligned Gaussians are inherently redundant and over-parameterized. This observation specifically motivated our design of a compact initialization model, which avoids this redundancy from the start.
> > > >
> > > > Regarding practicality and computational cost, we have demonstrated that our recurrent architecture can effectively refine approximately 500K Gaussians. More specifically, we have reported state-of-the-art results in this work with different numbers of input views (2, 8, 16, and 32) and different image resolutions (256x256, 256x448, and 540x960). This confirms that our method remains computationally tractable and memory-efficient, even when handling complex scenes with a large number of primitives.
> > > >
> > > > > The recurrent network takes a hidden state as input. From my understanding, the Gaussian parameters themselves evolve over iterations and could naturally serve as the hidden state, since they are updated and reused. Could the authors clarify why a separate hidden state is necessary and how it is initialized?
> > > >
> > > > Unlike the low-level, raw Gaussian parameters, the state encodes high-level image and 3D features extracted from our initialization model (it's initialized from our initial model and refined over iterations, see L190-205 of our manuscript). Such a feature-level input encodes more informative image and 3D context information compared to the raw 3D Gaussian parameters, which makes the learning more effective. The comparison is shown below and it's included in Table 6(d) of our revised manuscript.
> > > >
> > > > |           |   PSNR    |   SSIM    |   LPIPS   |
> > > > | --------- | :-------: | :-------: | :-------: |
> > > > | w/ state  | **29.07** | **0.902** | **0.105** |
> > > > | w/o state |   27.79   |   0.878   |   0.125   |
> > > >
> > > > > The paper reports strong generalization to unseen resolutions. I understand that the recurrent naturally allow adaptation to unseen data distributions (e.g., RealEstate10K) by iteratively refining Gaussians using rendering error, but it is unclear how this mechanism supports resolution generalization. Could the authors elaborate on this aspect?
> > > >
> > > > We thank the reviewer for this insightful question. We attribute this generalization capability to a fundamental structural difference: unlike previous single-step models that operate in image space (predicting Gaussian parameters per image), our recurrent model operates in global 3D space (processing a set of unorderd 3D points/Gaussians). While the rendering error is computed on the 2D image grid, it is propagated to the 3D points. Our recurrent network then processes these point-wise features independently of the image grid size. Consequently, the network learns a resolution-invariant update rule applied to 3D Gaussians, rather than a resolution-dependent regression over a fixed pixel lattice. This decouples the network's operations from the input/output resolution.

---

### Official Review · Reviewer_N2FZ · 2025-11-01

**Soundness:** 4
**Presentation:** 3
**Contribution:** 3
**Rating:** 6
**Confidence:** 4

**Summary:**

The authors propose ReSplat, a feed-forward recurrent Gaussian splatting model that iteratively refines 3D Gaussians without explicitly computing gradients. The authors find that rendering error in a *feature* space is an effective signal for learning Gaussian updates at every refinement iteration. The authors argue that the learned ReSplat model generalizes across different datasets and image resolutions via evaluations and ablations on the DL3DV and RealEstate10K datasets. To further speed up ReSplat, the authors find a quality-speed sweet-spot where, to initialize the recurrent process, the images are spatially downsampled by 16x, resulting in faster initialization and fewer Gaussians for the subsequent recurrent refinement step. This improves speed without a significant impact on image quality.

**Strengths:**

Predicting 3D Gaussians without explicit optimization offers a drastic speedup compared to optimization-based methods like 3D GS, Mip-Splatting, and Scaffold-GS.

Quantitative results show superior speed and image quality on both same- and cross-dataset evaluation on DL3DV and RealEstate10K when compared to concurrent methods.

Public release is a big plus for an application-focused paper
> (L308) We will release our code, pre-trained models, training and evaluation scripts to ease reproducibility.

**Weaknesses:**

I believe only training and testing on two datasets, namely DL3DV and RealEstate10K, does not completely validate the method's generalizability across datasets.

Additionally, since this is a neural-network-based approach, there remains important questions about generalization that have yet to be answered. What is the break-even point in terms of the number of training images at which ReSplat becomes better than 3D GS? If given more training views per scene, such as 32 or 64, would ReSplat still generalize and have better performance than 3D GS? Can global attention exploit information from many views even when it was only trained on a few? I would be happy to raise my rating if provided with additional ablations.

**Questions:**

It is possible to use feature spaces from newer, more modern vision models? E.g. DINOv2? I don't see a particular reason to stick with ResNets.

During training, what is "T", the number of times for which the RNN is rolled out?

Can you include GPU profiling for different parts of the model inference, such as a breakdown between the kNN attention and the global attention blocks? It would be insightful to see which parts uses the most wall-clock time and FLOPs, so perhaps a smarter trade-off between quality and compute can be found.

---

> ### Author Response · Authors · 2025-12-03
> **Official Response by Authors (1/3)**
>
> We thank the reviewer for the constructive feedback. We address the individual comments below.
>
> > I believe only training and testing on two datasets, namely DL3DV and RealEstate10K, does not completely validate the method's generalizability across datasets.
>
> We show the zero-shot cross-dataset generalization results below (RealEstate10K to ACID), where our ReSplat outperforms previous methods by 1dB PSNR, demonstrating the generalization of our method.
>
> These results are included in Table 4 of the revised manuscript.
>
> | Method     |  PSNR ↑   |  SSIM ↑   |  LPIPS ↓  |
> | :--------- | :-------: | :-------: | :-------: |
> | pixelSplat |   27.64   |   0.830   |   0.160   |
> | MVSplat    |   28.15   |   0.841   |   0.147   |
> | DepthSplat |   28.37   |   0.847   |   0.141   |
> | GS-LRM     |   28.84   |   0.849   |   0.146   |
> | ReSplat    | **29.87** | **0.864** | **0.135** |
>
> > Since this is a neural-network-based approach, there remains important questions about generalization that have yet to be answered. What is the break-even point in terms of the number of training images at which ReSplat becomes better than 3D GS? If given more training views per scene, such as 32 or 64, would ReSplat still generalize and have better performance than 3D GS?
>
> We show the comparisons with optimization-based 3DGS across 8, 16, and 32 views below. We outperform 3DGS by +2.8dB PSNR (8 views), +1.7dB (16 views), and +0.4dB (32 views). While the quality gap narrows at 32 views, ReSplat maintains superior performance. In addition, our ReSplat is $200 \times$ faster in terms of the reconstruction speed.
>
> These results are included in Table 7 of the revised manuscript and the discussions are added in L705-710.
>
> | # Views | Method  | Category     | # Iterations |  PSNR ↑   |  SSIM ↑   |  LPIPS ↓  | # Gaussians | Recon. Time (s) |
> | :-----: | :------ | :----------- | :----------: | :-------: | :-------: | :-------: | :---------: | :-------------: |
> |    8    | 3DGS    | Optimization |     4000     |   26.44   |   0.841   |   0.134   |    250K     |       49        |
> |    8    | ReSplat | Feed-Forward |    **4**     | **29.20** | **0.904** | **0.104** |   **57K**   |    **0.21**     |
> |   16    | 3DGS    | Optimization |     4000     |   27.38   |   0.864   |   0.119   |    395K     |       70        |
> |   16    | ReSplat | Feed-Forward |    **4**     | **29.01** | **0.900** | **0.105** |  **114K**   |    **0.34**     |
> |   32    | 3DGS    | Optimization |     4000     |   27.86   |   0.879   | **0.113** |    522K     |       160       |
> |   32    | ReSplat | Feed-Forward |    **4**     | **28.30** | **0.891** |   0.114   |  **229K**   |    **0.75**     |

---

> > ### Author Response · Authors · 2025-12-03
> > **Official Response by Authors (2/3)**
> >
> > > Can global attention exploit information from many views even when it was only trained on a few? I would be happy to raise my rating if provided with additional ablations.
> >
> > We evaluated our 8-view pre-trained initial and recurrent models with more than 8 input views below, and observed that our recurrent model benefits more from the additional input views, while the initial model tends to be saturated. This indicates that our rendering error-informed recurrent model exploits the additional information more effectively.
> >
> > These results are included in Figure 5(b) of the revised manuscript and the discussions are added in L416-419.
> >
> > From Figure 5 of the revised manuscript, we can see that our recurrent model generalizes robustly to unseen datasets, view counts and image resolutions.
> >
> > | **# Views** | **Method**              | **PSNR**                           | **SSIM**                            | **LPIPS**                           |
> > | ----------- | ----------------------- | ---------------------------------- | ----------------------------------- | ----------------------------------- |
> > | **8**       | ReSplat (init)          | 26.21                              | 0.842                               | 0.185                               |
> > |             | **ReSplat (recurrent)** | **27.70 *(+1.49)***                | **0.868 *(+0.026)***                | **0.160 *(-0.025)***                |
> > | **10**      | ReSplat (init)          | 26.61                              | 0.852                               | 0.176                               |
> > |             | **ReSplat (recurrent)** | **28.18 *(+1.57)*** | **0.877 *(+0.025)*** | **0.151 *(-0.025)*** |
> > | **12**      | ReSplat (init)          | 26.77                              | 0.857                               | 0.171                               |
> > |             | **ReSplat (recurrent)** | **28.47 *(+1.70)*** | **0.883 *(+0.026)*** | **0.144 *(-0.027)*** |
> > | **14**      | ReSplat (init)          | 26.88                              | 0.862                               | 0.166                               |
> > |             | **ReSplat (recurrent)** | **28.75 *(+1.87)*** | **0.888 *(+0.026)*** | **0.139 *(-0.027)*** |
> > | **16**      | ReSplat (init)          | 26.90                              | 0.865                               | 0.164                               |
> > |             | **ReSplat (recurrent)** | **28.92 *(+2.02)*** | **0.892 *(+0.027)*** | **0.135 *(-0.029)*** |
> >
> >
> >
> > > It is possible to use feature spaces from newer, more modern vision models? E.g. DINOv2? I don't see a particular reason to stick with ResNets.
> >
> > We compare ResNet features with those from DINOv2. We observed no improvement when using the larger, more recent feature extractor. We attribute this to the patch-based architecture of DINOv2, which may result in coarser spatial information. In contrast, convolutional networks maintain local structural fidelity, which is critical for high-quality pixel-accurate view synthesis. In addition, ResNet runs much faster than DINOv2, making our model more efficient.
> >
> > These results are included in Table 12 of the revised manuscript and the discussions are added in L749-753.
> >
> > | Features   | #Parameters | PSNR ↑ | SSIM ↑ | LPIPS ↓ |
> > | :--------- | :---------: | :----: | :----: | :-----: |
> > | **ResNet** |    0.7M     | 29.07  | 0.902  |  0.105  |
> > | **DINOv2** |    86.6M    | 29.00  | 0.901  |  0.107  |
> >
> > > During training, what is "T", the number of times for which the RNN is rolled out?
> >
> > During training, we randomly sample the number of iterations $T$ between 1 and 4, and our model supports different number of iterations at inference time, allowing a flexible trade-off between accuracy and speed with a single model.
> >
> > This is clarified in L259-L261 of the revised manuscript.

---

> > > ### Author Response · Authors · 2025-12-03
> > > **Official Response by Authors (3/3)**
> > >
> > > > Can you include GPU profiling for different parts of the model inference, such as a breakdown between the kNN attention and the global attention blocks? It would be insightful to see which parts uses the most wall-clock time and FLOPs, so perhaps a smarter trade-off between quality and compute can be found.
> > >
> > > We thank the reviewer for the suggestion. Below we report the total runtime and individual component latency of both the initial reconstruction model and the recurrent model. In the initial reconstruction model, the depth prediction module constitutes the majority of the runtime. For the recurrent model, the kNN attention mechanism consumes the most time. These results highlight potential areas for future optimization.
> > >
> > > The results are included in Table 13 of the revised manuscript and the discussions are added in L755.
> > >
> > > Table: Initial Model Profiling
> > >
> > > | **Resolution** | **Total** | **Depth pred.** | **kNN attn** | **Global attn** | **Gaussian head** |
> > > | -------------- | --------- | --------------- | ------------ | --------------- | ----------------- |
> > > | 256 × 448      | 0.149     | 0.111           | 0.024        | 0.013           | 0.001             |
> > > | 512 × 960      | 0.311     | 0.197           | 0.094        | 0.018           | 0.002             |
> > >
> > > Table: Recurrent Model Profiling
> > >
> > > | **Resolution** | **Total** | **Render error** | **kNN attn** | **Global attn** | **Update head** |
> > > | -------------- | --------- | ---------------- | ------------ | --------------- | --------------- |
> > > | 256 × 448      | 0.022     | 0.003            | 0.015        | 0.002           | 0.002           |
> > > | 512 × 960      | 0.126     | 0.016            | 0.092        | 0.008           | 0.010           |

---

### Author Response · Authors · 2025-12-03
**General response**

We thank all reviewers for their constructive feedback and insightful comments. Below, we summarize our responses to the major questions raised.

### **1. More Comparisons with 3DGS & Break-even Point (Reviewer N2FZ)**

Reviewer N2FZ asked about the performance break-even point regarding the number of training images. We compared ReSplat with optimization-based 3DGS across 8, 16, and 32 input views.

- **Quality**: We outperform 3DGS by **+2.8dB PSNR (8 views)**, **+1.7dB (16 views)**, and **+0.4dB (32 views)**. While the quality gap narrows at 32 views, ReSplat maintains superior performance.

- **Speed**: ReSplat is **200x faster** in reconstruction speed. Our method completes reconstruction in less than **0.8 seconds (zero-shot)**, whereas optimization-based 3DGS requires more than **2 minutes per scene**.

These results are included in Table 7 of the revised manuscript.

### **2. Optimization-based vs. Feed-Forward Refinement (Reviewer 45AN)**

Reviewer 45AN requested to compare optimization-based refinement with our feed-forward refinement by using the same ReSplat's feed-forward initialization.

- **Convergence**: ReSplat requires only **4 iterations** to achieve the same quality that optimization-based 3DGS achieves in **80 iterations**.

- **Speed**: Consequently, ReSplat is **53x faster** in reconstruction speed thanks to the significantly reduced number of iterations. These results highlight the efficiency and high quality of our feed-forward refinement.

These results are included in Figure 4 and Table 8 of the revised manuscript.

### **3. Recurrent vs. Non-recurrent Network (Reviewers 45AN, ZHGA)**

Reviewer 45AN and ZHGA raised the question of whether the recurrent design is essential. We compared with non-recurrent variants.

- **Non-weight-sharing (Multi-step)**: Using different weights for each iteration (while keeping other components identical) increases parameters by **4x** and degrades performance by **-0.3dB PSNR**.

- **Single-step Deeper Network**: Increasing attention blocks for a single-step refinement results in **4x** more parameters and a **-0.7dB PSNR** drop compared to our approach.

- **Conclusion**: The weight-sharing design acts as an implicit regularization, making the network more parameter-efficient while achieving superior synthesis results.

These results are included in Table 9 of our revised manuscript.

### **4. Generalization Capabilities (Reviewers N2FZ, ZHGA)**

- **Cross-Dataset**: In a zero-shot transfer from RealEstate10K to ACID, ReSplat outperforms previous methods by **+1.0dB PSNR**.

- **Robustness**: We evaluated generalization across unseen datasets, varying input view counts, and unseen image resolutions. ReSplat generalizes significantly better than prior methods, as the rendering error serves as an effective feedback signal for adapting to diverse test scenarios.

These results demonstrate that **our recurrent model generalizes robustly to unseen datasets, view counts and image resolutions**, and they are included in Table 4 and Figure 5 of the revised manuscript.

We have resolved all other specific questions in the individual reviewer responses and incorporated these additional experiments and discussions into the revised manuscript. Thank you again for your time and effort.

---

### Meta-Review · Area_Chair_AEDj · 2026-01-05

**Summary:**

The reviewers recognized ReSplat's motivation to refine feed-forward 3DGS iteratively, noting its significant 200× speedup over optimization-based methods. However, consensus remains that the method functions more as an effective initialization for subsequent 3DGS optimization rather than a standalone reconstruction pipeline. Additionally, reviewers highlighted that the proposed iterative process is slower than competing feed-forward methods. The manuscript received an average score of 4.67, and none of the reviewers has replied in the discussion phase.

**Reviewer Concerns:**

Several concerns might have been addressed by the rebuttal:
- Generalization: the authors have provided zero-shot results on the ACID dataset, which might demonstrate the model's ability to adapt to unseen data.
- Empirical benchmarking: table 7 clarified the break-even point, demonstrating that ReSplat maintains a lead over standard 3DGS in sparse-view settings (up to 32 views).
- Architecture validation: ablations confirmed the parameter efficiency of the weight-sharing recurrent design over deeper single-step models.

However, the following concern might still be valid:
- Recurrent method or just better initialization: the overall method is still similar to feedforward initialization and dedicated strategies for 3DGS optimization.

**Reviewer Scores:**

The manuscript received an initial score of (6, 4, 4) and none of the reviewers replied during the discussion phase. Given the equivalent initialization concern, I am not sure whether reviewers 45AN and ZHGA would raise their scores. Thus, I would assume a final score of 4.5-5.5. In the absence of a strong champion to advocate for the paper’s conceptual novelty, I cannot support the paper at the current stage.

---

### Decision · Program_Chairs · 2026-01-26

Reject